# Spatial and Temporal Variability of Dense Shelf Water Cascades along the Rottnest Continental Shelf in Southwest Australia

**Tanziha Mahjabin \*** [ID], **Charitha Pattiaratchi** [ID], **Yasha Hetzel** [ID] **and Ivica Janekovic** [ID]

Oceans Graduate School & the UWA Oceans Institute, The University of Western Australia, 6009 Crawley, Australia; chari.pattiaratchi@uwa.edu.au (C.P.); yasha.hetzel@uwa.edu.au (Y.H.); ivica.janekovic@uwa.edu.au (I.J.)
**\*** Correspondence: tanziha.mahjabin@research.uwa.edu.au; Tel.: +61-410-294437

**Abstract:** Along the majority of Australian shallow coastal regions, summer evaporation increases the salinity of shallow waters, and subsequently in autumn/winter, the nearshore waters become cooler due to heat loss. This results in the formation of horizontal density gradients with density increasing toward the coast that generates gravity currents known as dense shelf water cascades (DSWCs) flowing offshore along the sea bed. DSWCs play important role in ecological and biogeochemical processes in Australian waters through the transport of dissolved and suspended materials offshore. In this study a numerical ocean circulation model of Rottnest continental shelf, validated using simultaneous ocean glider and mooring data, indicated that the passage of cold fronts associated with winter storms resulted in rapid heat loss through evaporative cooling. These conditions resulted in enhancement of the DSWCs due to modifications of the cross-shelf density gradient and wind effects. Specifically, onshore (offshore) directed winds resulted in an enhancement (inhibition) of DSWCs due to downwelling (vertical mixing). Consequently, the largest DSWC events occurred during the cold fronts when atmospheric temperatures reinforced density gradients and onshore winds promoted downwelling that enhanced DSWCs. Advection of DSWCs was also strongly influenced by the wind conditions, with significantly more transport occurring along-shelf compared to cross-shelf.

**Keywords:** dense shelf water cascades; cross-shelf exchange; wind effects; numerical modelling; Rottnest continental shelf; southwest Australia

## 1. Introduction

Australia has a high rate of evaporation, around 2.5 m per year [1] with low rainfall and river run-off that generally results in coastal waters having higher salinity than offshore. Along the majority of Australian shallow coastal regions, summer evaporation leaves the shallow coastal waters more saline and subsequently in autumn and winter the nearshore waters become cooler due to heat loss via convection [2,3]. Combination of salinity and cooling effects causes strong horizontal density gradients to develop with density increasing from the ocean towards the coast. This horizontal density gradient is the driving force for the formation of buoyancy-driven flows along the sea bed, defined as dense shelf water cascades (DSWCs) [4–7]. DSWC have important ecological and biological implications as they provide an effective mechanism to transport nearshore water and dissolved and suspended material (e.g., terrestrial carbon, nutrients, larvae, low-oxygen water, sediments, and pollutants) off the continental shelves into the deep ocean [3]. Despite their ecological importance, DSWCs are rarely measured in detail because the process often consists of intermittent events occurring in the bottom layers that cannot be observed using satellite measurements [6]. DSWCs have, however, been documented globally [8] and around the Australian coastline using

hydrographic data collected along single or repeated cross-shelf transects measured using ship or ocean gliders [2]. These benthic flows are identifiable in cross-shelf density profiles as a wedge of denser (colder, saltier, or a combination) water with the nose of the wedge protruding down the shelf under the less dense offshore water [2]. However, the spatial/temporal characteristics and controlling forces vary broadly, and understanding DSWC dynamics has been limited and based on many assumptions [6].

The underlying driving force for DSWCs is the horizontal density gradient with inhibiting effects, mostly via vertical mixing resulting from either wind and/or tidal mixing [2,5,9,10]. In the majority of sites, data indicated that when wind and tidal mixing were weak, either a bottom gravity current or surface plume would form, or when vertical mixing was strong, the water column was well mixed [2,3]. On the continental shelf, numerous processes play important roles including, for example, the effects of: boundary currents, advection, eddies, topography, wind-driven currents, downwelling and upwelling, and ambient density fields [6].

In Australia, shipborne surveys focusing on DSWCs have been conducted on the Northwest Australian Shelf [7,11], Shark Bay [12], South Australian Gulfs [13], The Great Australian Bight [14], Hervey Bay [15], and Jervis Bay [16]. In these cases, the DSWCs were either described in detail in semi-enclosed bays [10,17], or by sparse sampling on the continental shelf.

More recently, high-resolution ocean glider data have provided evidence that DSWCs are common features that occur all around the Australian coastline, even where winds and tidal currents are strong [2]. In southwest Australia, glider data collected over many years along a repeated cross-shelf transect showed that DSWCs were present a majority of the time during autumn and winter when mean wind speeds were low [5]. These authors hypothesized that this buoyancy-driven process transported a significant portion of dense shelf water and suspended matter across the shelf. Although the gliders provided monthly snapshots of cascade structure along a single transect, the temporal evolution, flow pathways, velocities, occurrence outside of the study site, and the effects of passing weather systems on the cascades were not defined. More recently, glider and mooring observations of velocity and vertical stratification at one site [2] indicated that weather conditions, in particular the wind direction, strongly influenced DSWCs. Understanding of the behavior of DSWCs at these shorter time scales (hours to days), and their spatial distribution is important for predicting the occurrence of individual events along the coast as well as the cumulative impacts.

Therefore, the main aim of this paper is to focus on the spatial and temporal variability along the Rottnest continental shelf (Figure 1). In particular, to define (1) the spatial extent and pathways of DSWCs; and (2) the influence of the changing synoptic weather conditions. To achieve this, a numerical circulation model, validated using an ocean glider and current measurements, was applied to the study region (Figure 1). The period May–June 2016, representing late austral autumn and early winter period, was selected for the study as previous observational studies identified the frequent occurrence of DSWC [2,5].

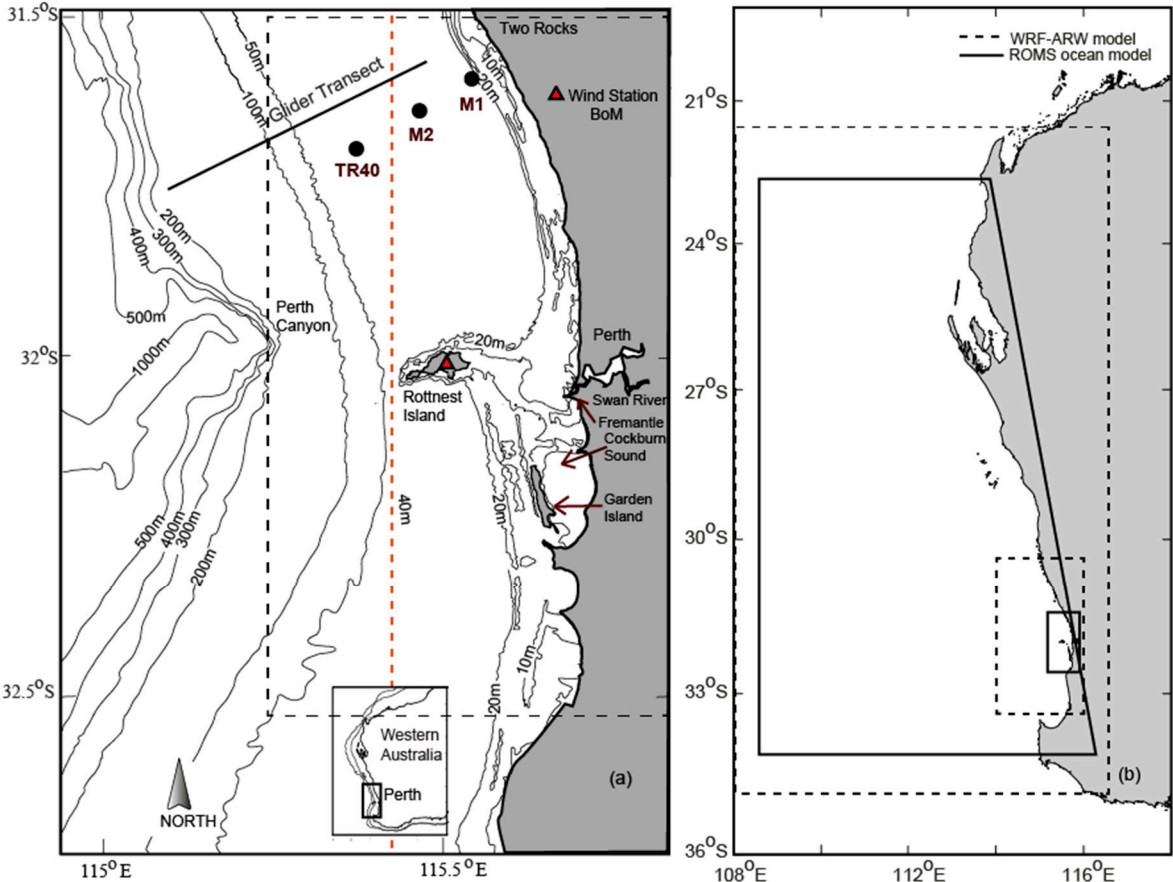

**Figure 1.** (**a**) Location of glider transect and mooring deployments (M1, M2, TR40) in May 2016 with the bathymetry of Two Rocks continental shelf. Wind data was used from the station situated at Rottnest Island maintained by the Bureau of Meteorology. The black dashed line represents the hydrodynamic nested model domain and red dashed line represents the sections shown on Figures 9 and 13 close to a 40 m water depth. (**b**) Large scale parent grid and nested child domain for both the Regional Ocean Modelling System (ROMS) ocean modeling system and Weather Research and Forecasting Advanced Research (WRF-ARW) model.

This paper is arranged as follows: Section 1.1 gives an overview of the study area, Section 2 details the hydrodynamic model setup. Results and discussion are given in Sections 3 and 4, respectively, with conclusions in Section 5.

## 1.1. Study Site

This study is focused on the Rottnest continental shelf (RCS) (depth < 50 m) in southwestern Australia (Figure 1). The bathymetric features of this region include (Figure 1): (1) the shallow inshore regions where the depths are <10 m having discontinuous submerged limestone reefs; (2) the upper continental shelf from around ≈10 km to ≈40 km having a mean depth of 40 m; (3) the depth increases rapidly in the lower continental shelf between the 50 m and 100 m isobaths; (4) the shelf break located at the ≈200 m isobath [5]; (5) the presence of Rottnest Island that interrupts the shore parallel flow; (6) the Perth canyon located to the west of Rottnest Island; and (7) Cockburn Sound, a semi-enclosed coastal embayment.

The diurnal micro tidal (tidal range ≈0.5 m) [5] mean that the tidal currents have a negligible influence in the study region [18,19]. The main freshwater input to the system is through the Swan River at Fremantle (Figure 1). Although the river has a large catchment the discharge is relatively small and results in only a localized region of freshwater influence.

The climate may be classified as "Mediterranean" with annual mean evaporation rates and rainfall of ≈2.5 m and ≈730 mm, respectively [20]. During the summer, coastal heating and evaporation result in a band of warmer, higher salinity water in the coastal boundary layer [21], which establishes a horizontal density gradient that is reinforced when cooling occurs during autumn and winter. This cross-shelf density gradient is crucial to the cross-shelf transport over the inner continental shelf [22,23]. May and June are typically when the horizontal density gradient reaches its maximum and DSWCs have been observed frequently during this time period [2,5].

Continental shelf processes in the study region are mainly wind driven, with three wind systems dominating: sea breezes, storms (wind speeds >17 m/s), and calm periods (wind speeds <7 m/s) [24]. Local sea breezes, superimposed upon synoptic southerly winds (with speeds often >15 m/s), are prevalent in austral summer and spring (September–February) [25]. Storm systems are most frequent during late autumn and winter (May–August), and with the passage of frontal systems with maximum wind speeds > 30 m/s. On average, up to 30 storms per year will impact on the study region [26]. Winter storms have a typical pattern with strong north/northeasterly winds blowing for 12 to 52 h, followed by a period of similar duration when winds turn south/southwesterly, with no prevailing direction dominating for the duration of the storm. Calm wind conditions are mainly observed between the passage of winter storm fronts and are characterized by low wind speeds (<5 m/s). After the passage of the cold front, winds rotate to the southwest resulting in an onshore component that contributes to downwelling conditions [27].

The study region is influenced by the Leeuwin and Capes Currents [28]. The poleward flow of the Leeuwin Current (LC) transports warmer lower salinity water along the 200 m depth contour [12,29,30]. The LC flows strongest during the austral autumn/winter (April–September) when the opposing southerly winds and sea breezes are weakest [12,31]. The Capes Current (CC) is a seasonal inner shelf wind-driven current, flows northward inshore of the LC during summer months when southerly wind (>7 m/s) and sea breezes prevail [32,33]. The study is also located in a region where the local inertial period (22.6 h) is close to 24 h and thus a resonance condition between the local sea breeze system and inertial currents occur, particularly during the summer months when strong prominent sea breeze cycles occur [19].

## 2. Methodology

### 2.1. Numerical Models

The spatial and temporal variability of DSWCs were investigated through analysis of Regional Ocean Modelling System (ROMS) output archived (2015–present) from a real time forecast system for the west coast of Western Australia developed at The University of Western Australia (www. coastaloceanography.org). The analysis presented here is focused on the period May–June 2016 for which coinciding ocean glider and oceanographic mooring data for model validation were available.

ROMS is a 3-D hydrostatic, non-linear, free surface, s-coordinate, time-splitting, finite-difference primitive equation numerical ocean model [34,35]. A more detailed description of the model and numerical schemes can be found on the ROMS webpage (http://www.myroms.org).

The modelling system consisted of two one-way nested curvilinear grid domains (Figure 1b): (i) larger scale "parent" grid for the whole of the western region of Australia at ≈1.5–2.5 km horizontal resolution (700 × 1320 km), and an embedded (ii) nested "child" domain for the wider Perth region including the Rottnest continental shelf at a 500 m resolution. Bathymetry data were derived from a combination of: The General Bathymetric Chart of the Oceans (GEBCO) (http://www.gebco.net/data_and_products/gridded_bathymetry_data/), Geoscience Australia 250m [36], and high-resolution LIDAR data in the shallow coastal regions. Final bathymetry was minimally smoothed using a linear programming approach [37] to suppress horizontal pressure gradient errors. In the vertical, 20 sigma layers were used for the child domain. Sigma coordinates allowed for more layers in the bottom layer to resolve the DSWCs.

Non-tidal parent domain boundary conditions for the free surface, temperature, salinity, and velocity were taken from the global 1/12 degree Hybrid Coordinate Ocean Model (HYCOM) model (http://hycom.org) and combined with nine tidal constituents for elevation and barotropic velocities from the global barotropic tidal inverse solution (TPXO) [38]. Nested child model boundaries were updated every 600 s, downscaling dynamics from the parent model. Barotropic boundary velocities were prescribed using the Flather scheme [39], and for baroclinic velocity and tracers (temperature and salinity), a combination of Orlanski-type radiation boundary conditions with nudging [40] were used. The advection scheme used for active tracers (temperature and salinity) in the child domain was a multidimensional positive definite advection transport algorithm (MPDATA) [41], which was a conservative and positive definite scheme, while for the parent system, the upwind third-order scheme was used [42]. For baroclinic momentum, the upwind third-order advection scheme for both models was used. Generic length scale [43] was used for turbulent closure.

As shown in other similar studies [44–47], this configuration included non-linear dynamics, whilst the hydrostatic approximation used is not a major limitation because the horizontal scales of motion are larger than the vertical ones, and effects have been shown to be limited to plume head dynamics for extreme cases [48], which is not the focus here.

Atmospheric forcing was prescribed every hour via bulk formulation [49] using all required variables from a locally tuned atmospheric modelling system based on the WRF-ARW model core (also available on the University of Western Australia (UWA) coastal oceanography website: www.coastaloceanography.org). Similar to the ROMS model grids, the WRF-ARW model was also configured for two domains (Figure 1b), covering a region slightly larger than those for the ROMS model, using a two-way coupled mode for boundary information exchange. The atmospheric parent model domain had a 10 km horizontal resolution and 45 levels in the vertical, completely covering Western Australia. The ROMS parent domain with a nested domain for the wider Perth region was defined at a 2 km horizontal resolution. Boundary and initial conditions for the parent atmospheric model were interpolated from the global Naval Oceanographic Office Global Navy Coastal Ocean Model (NCOM) Global Forecast System (GFS) model at 0.25° resolution. Both atmosphere and ocean model outputs were saved hourly and served using the Open-source Project for a Network Data Access Protocol openDAP thredds server (http://130.95.29.59:8080/thredds/catalog.html).

### 2.2. Field Observations

We used high-resolution temperature, salinity, and density data from Teledyne Webb Research Slocum Electric Gliders (Falmouth, MA, USA) [50] that were deployed at Two Rocks in May 2016 to repeat cross-shelf transects and verify the ability of the model to reproduce DSWCs at this site. All gliders were operated by the ocean glider facility located at the University of Western Australia with data available through the Australian Ocean Data Network (https://portal.aodn.org.au). The Slocum Gliders for this experiment collected data (temperature, salinity, density, fluorescence, suspended sediment, and dissolved oxygen) from the surface to seabed to a maximum depth of 200 m traversing at a mean speed of 25 km per day [50]. Gliders use buoyancy control whilst moving forward to the target destination to navigate their way to a series of pre-programmed waypoints using GPS, internal dead reckoning, and altimeter measurements. The gliders were equipped with a Sea-Bird Scientific (Bellevue, WA, USA) pumped CTD (conductivity–temperature–depth) sensor, a WETLabs (Bellevue, WA, USA) 3 parameter optical sensor (which measured chlorophyll fluorescence, coloured dissolved organic matter and backscatter at 660 nm) and an Aanderaa (Nesttun, Norway) dissolved oxygen Optode [50]. All the sensors were sampled at 4 Hz (which yielded measurements ~7 cm in the vertical). However, only CTD data are presented in this paper.

We also used velocity and temperature data from the IMOS TR40 mooring (located at 31.72° S and 115.398° E) at a 40 m depth (Figure 1a) to validate the predicted current velocities. The shelf mooring consisted an upward looking RDI Workhorse Sentinel 600 acoustic Doppler current profiler (ADCP)

(Teledyne RDI, Poway, CA, USA) operating at a frequency of 600 kHz and collected velocity profiles in 2 m bins over the water column at 20 min intervals.

Satellite sea surface temperatures at a 2 km resolution were obtained from the gridded multi-swath IMOS L3S Advanced Very High Resolution Radiometer (AVHRR) dataset [51]. Sea Surface Temperature (SST) data were used both to verify surface temperatures and overall spatial patterns evident in the model. Clouds were present for most of May, with cloud-free data available only near the start and end of the analysis period.

Wind speed and wind direction, at 30 min intervals, were obtained from the Bureau of Meteorology weather station at Rottnest Island (Figure 1a).

### 2.3. Data Analysis

The bottom temperature was used as a proxy for DSWCs with the bottom sigma layer from the ROMS model used to illustrate flow pathways. Temporal means were calculated over periods of similar atmospheric conditions (~days) with wind speed and direction determining specific "events". This allowed for the definition of flow pathways during "weak" and "strong" wind events. East and north velocity components (for both model and observations) were considered analogous to cross-shelf and along-shelf directions, respectively. Heat flux data archived from the ROMS/WRF modelling system were used to assess the cooling over the two-month study period.

The time base for results presented here is UTC with the local time +8 h.

## 3. Results

### 3.1. Oceanographic and Meteorological Conditions

During May 2016, the study region (RCS) experienced a range of wind conditions with the west to east passage of extratropical low/high pressure systems at 7–10 days intervals. Strongest winds (>10 ms$^{-1}$) were associated with the passage of low pressure systems (storms) through the study area, whilst during the intervening periods, weaker winds were present (Figure 2a,b). Three strong storms occurred on May 8, 21, and 24, with maximum wind speeds (from the northwest) of 18, 24, and 20 ms$^{-1}$, respectively, at landfall of the storm. In the following 2–3 days, wind speeds decreased and rotated anticlockwise toward the west and southwest. Between storms, winds were generally below 10 ms$^{-1}$ and variable directions were observed (Figure 2a,b). The analysis presented here focuses on two representative periods: (1) weak winds for May 13–16, and (2) strong onshore winds associated with a strong winter cold front for May 20–23. Other periods were also briefly considered.

Time series of net surface heat flux was calculated from the bulk formula (and used for ROMS model forcing) at two locations (Figure 1a): (1) close to the coast (M2), and (2) ≈10 km offshore (M1), both indicated diurnal heating/cooling with daily amplitude of ≈200–400 Wm$^{-2}$ and maximum negative rates of −800 Wm$^{-2}$ (Figure 2c). Cumulative net heat loss (cooling) of 0.9–1.3 × 10$^{10}$ Jm$^{-2}$ occurred over the month (Figure 2d). The heat flux data indicated periods of rapid cooling associated with the passage of the cold fronts, with two notable events around May 6 and 20 (Figure 2d). These two periods of rapid heat loss over four days accounted for ≈35% of the total heat loss over the entire month.

Satellite sea surface temperature images (Figure 3a,b) illustrated the general cooling trend over the period and the enhancement of the band of cooler water near the coast. Offshore in the core of the Leeuwin Current, the SST decreased by ≈0.5 °C and near the coast temperatures decreased by >1 °C over two weeks (Figure 3), resulting in a strengthening of the cross-shelf density gradient. By the start of June, the entire inner shelf region had temperatures ≈19 °C (Figure 3b).

### 3.2. Model Validation

Model validation was focused on the ability of the model to reproduce DSWCs that have been observed in the region in cross-shelf transects measured by ocean gliders [2]. The model domain

included the glider trajectory on the continental shelf, although the offshore extent of the glider track was outside the model boundary (Figure 1). Both the model output and glider measurements indicated the occurrence of DSWC with similar patterns of temperature, salinity, and density (Figure 4). The estimated horizontal density gradient from the model output was $-18.4 \times 10^{-6}$ kg m$^{-4}$, close to the measured value of $-21.8 \times 10^{-6}$ kg m$^{-4}$. The model output also indicated that both temperature and salinity contributed to the distribution of density; however, over this period, temperature was the dominant contributor (Figure 4). Also, the range of model estimates of temperature, salinity, and density were in good agreement with the glider measurements, with a bias of 0.8 °C in temperature. Overall, similarities in the DSWC structure, location, and occurrence provided confidence that the model could be used as a tool to determine the spatial and temporal variability of DSWCs within the study region.

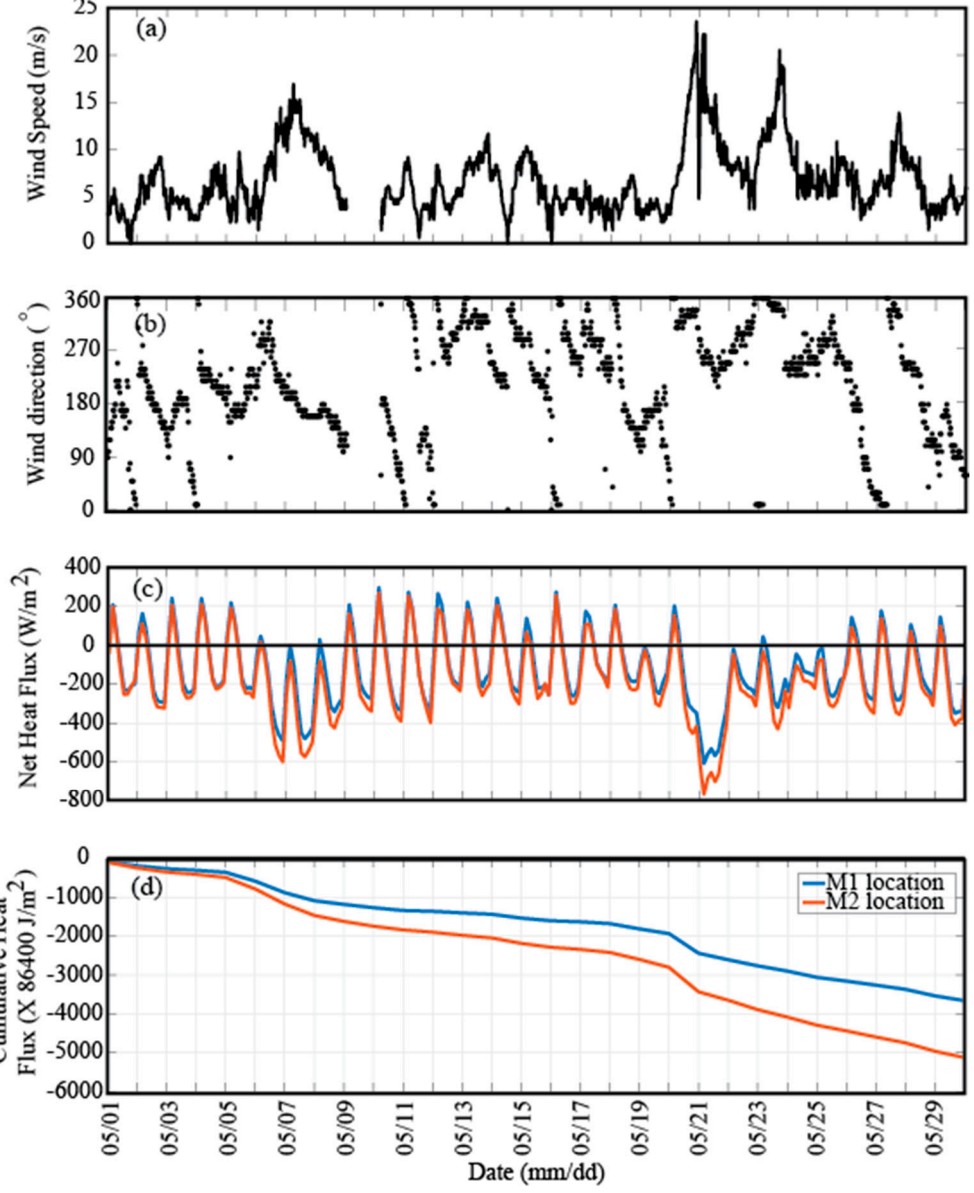

**Figure 2.** Time series of: (**a**) wind speed, (**b**) wind direction, (**c**) net heat flux, and (**d**) cumulative heat flux during May 2016. A negative value in net heat flux defined an upward flux or cooling, and a positive value defined a downward flux or heating. Note that the tick marks on the time axis are at 4 a.m. (UTC) corresponding to 12 noon (local time).

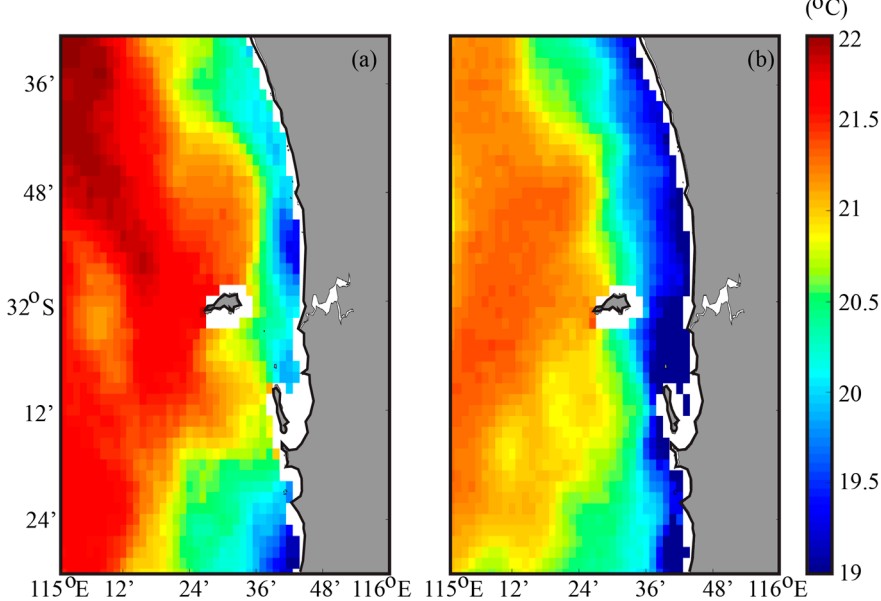

**Figure 3.** Satellite-derived 3-day mean sea surface temperature: (**a**) May 9–11, 2016, before the major storm event; and (**b**) at the end of the study period for June 1–3, 2016. Note the overall cooler temperatures and narrow band of cooler waters near the coast.

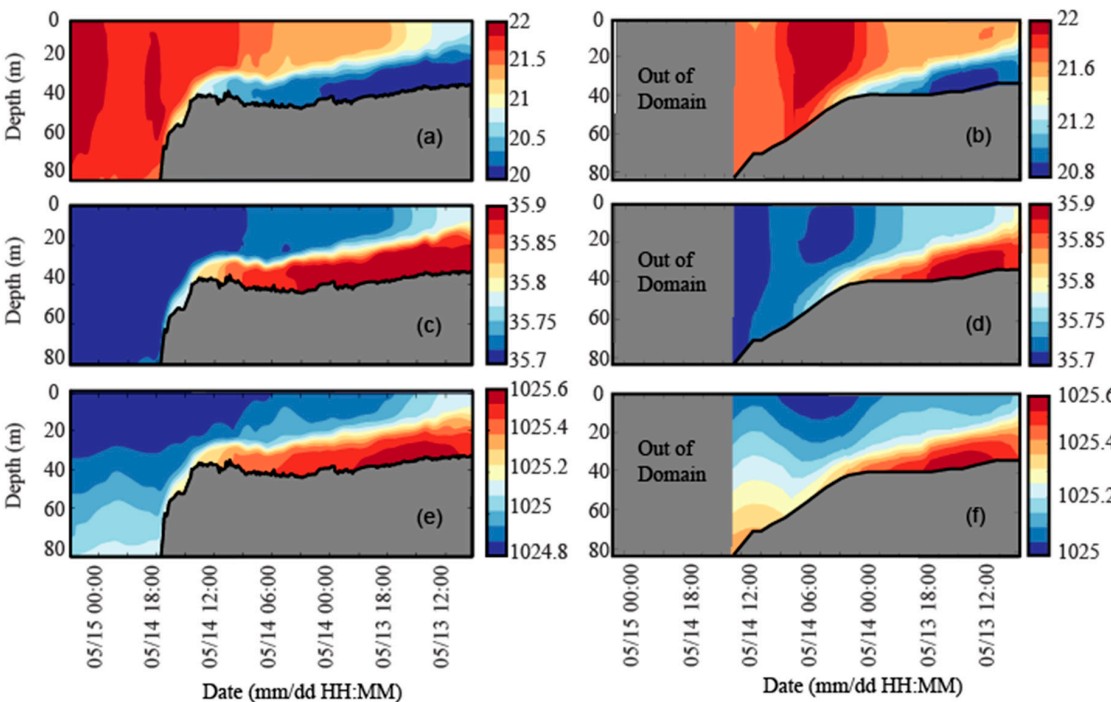

**Figure 4.** Comparison between observed (**a**) temperature (°C), (**c**) salinity (psu), and (**e**) density (kg/m³) distribution along the Two Rocks transect obtained from ocean glider data and (**b**,**d**,**f**) those predicted from ROMS simulation over May 13–15, 2016.

Current velocities were compared to those obtained using an ADCP located at TR40 mooring at a 40 m depth offshore of Two Rocks (Figure 1a). Velocity magnitudes were generally in good agreement (Figure 5). The main similarity was stronger for along-shelf flow (north/south) compared to cross-shelf (east/west), which oscillated between northerly and southerly flows persisting for 2–3 days at a time (Figure 5). The reversal of along-shelf flows was reflected in changes in wind conditions. The current

velocities were uniform through the water column during periods of stronger winds (e.g., May 20–23), whilst during periods of weak winds (e.g., May 13–19), there were changes in the velocity structure through the water column (Figure 5). During periods of vertical shear, near-bottom velocities tended to flow toward the south and were associated with DSWCs in both the model and observed data, adding more confidence in the model to reproduce the cascade dynamics. Between May 17–19, surface currents were toward the north whilst bottom currents flowed to the south (Figure 5). Mean along-shore flow profiles over the weak and strong wind events (Figure 5) indicated southward flow.

In the cross-shelf direction, observed and predicted velocities both indicated that onshore surface flow frequently overlaid offshore flow at depth (Figure 5). Velocity structure was generally more complex in the higher resolution observations, but the occurrence of pulses of near bed offshore flow associated with DSWCs had a similar structure, duration, and magnitude ($\approx$0.2 ms$^{-1}$), for example, around May 20 (Figure 5). Mean velocity profiles over the three wind events indicated vertical flow reversal, with onshore flow at the surface and offshore flow near the bottom that were associated with DSWCs (Figure 5).

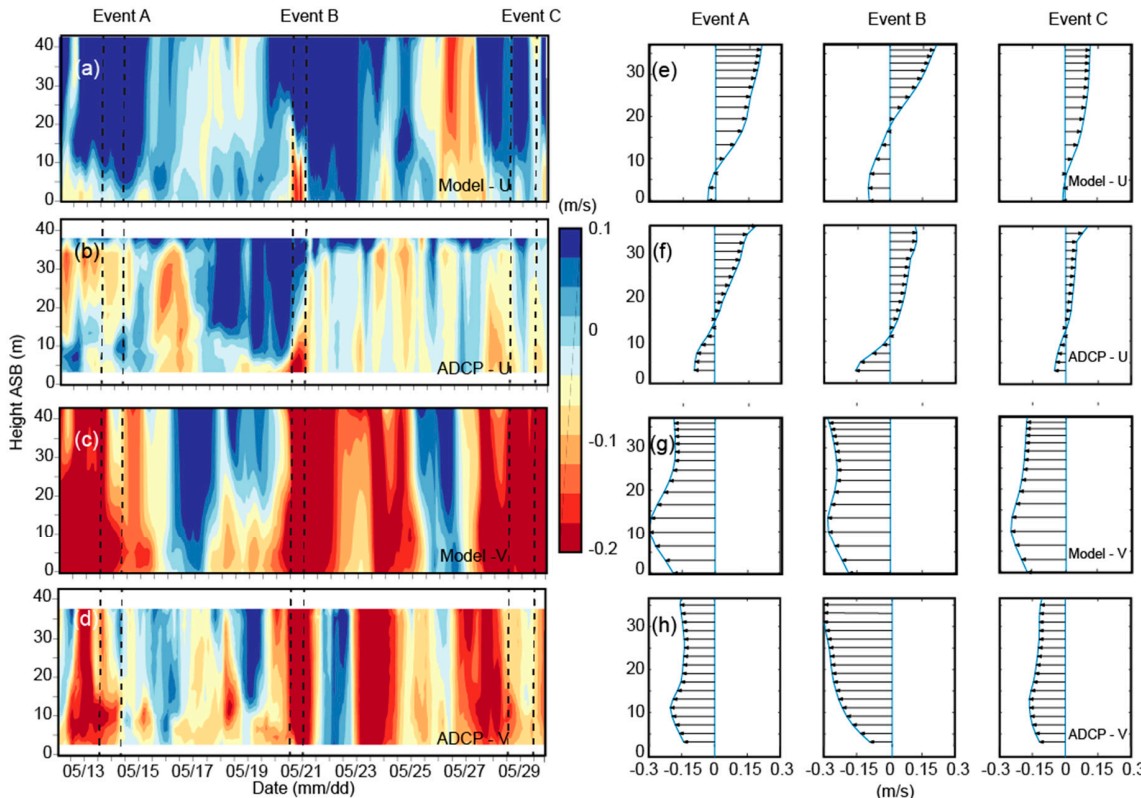

**Figure 5.** Time series (May 12–30, 2016) of (**a**) the predicted u-component of velocity, (**b**) observed u-component of velocity from ADCPs, (**c**) as for (**a**) for v-component velocities, (**d**) as for (**b**) for v-component velocities, (**e**) mean predicted velocity profile for each event for the u-component, (**f**) mean observed velocity profile for each event for the u-component, (**g**) as for (**e**) for v-component velocities, and (**h**) as for (**f**) for v-component velocities. Warmer colors represent flow to the west (u-component) and to the south (v-component). Dashed lines in panels a–d represent averaging periods for events in panels e–h.

Predicted cross-shelf temperature gradients and the magnitude of the cooling trend over May was also in close agreement with satellite SST values (Figure 6 vs. Figure 3). Since the horizontal temperature (density) gradient is the critical driving force for DSWCs, this suggested that the model not only reproduced instantaneous features but could also be used to look at DSWCs over seasonal or annual time scales, although that is not the intention here.

### 3.3. Occurrence of DSWC during Early Winter (May–June 2016)

Over May and June 2016, the predicted cross-shelf temperature sections along RCS indicated a general cooling trend, persistent horizontal temperature (density) gradients, and the frequent occurrence of DSWC (Figure 6). Offshore temperatures dropped ≈3 °C (from 23 °C to 20 °C), whilst nearshore temperatures reduced from ≈21.5 °C to <19.5 °C (Figure 6). Although the DSWC did not always reach the edge of the continental shelf, the wedge-shape structure of the cascades and associated vertical stratification was a common feature. In particular, strong DSWC events occurred on May 14, 22, and 28; and June 4, 12, and 18 (Figure 6). These events coincided with both strong and weak wind events, although the more intense DSWC events appeared to be associated with strong onshore winds (from the NW) (Figures 3 and 6).

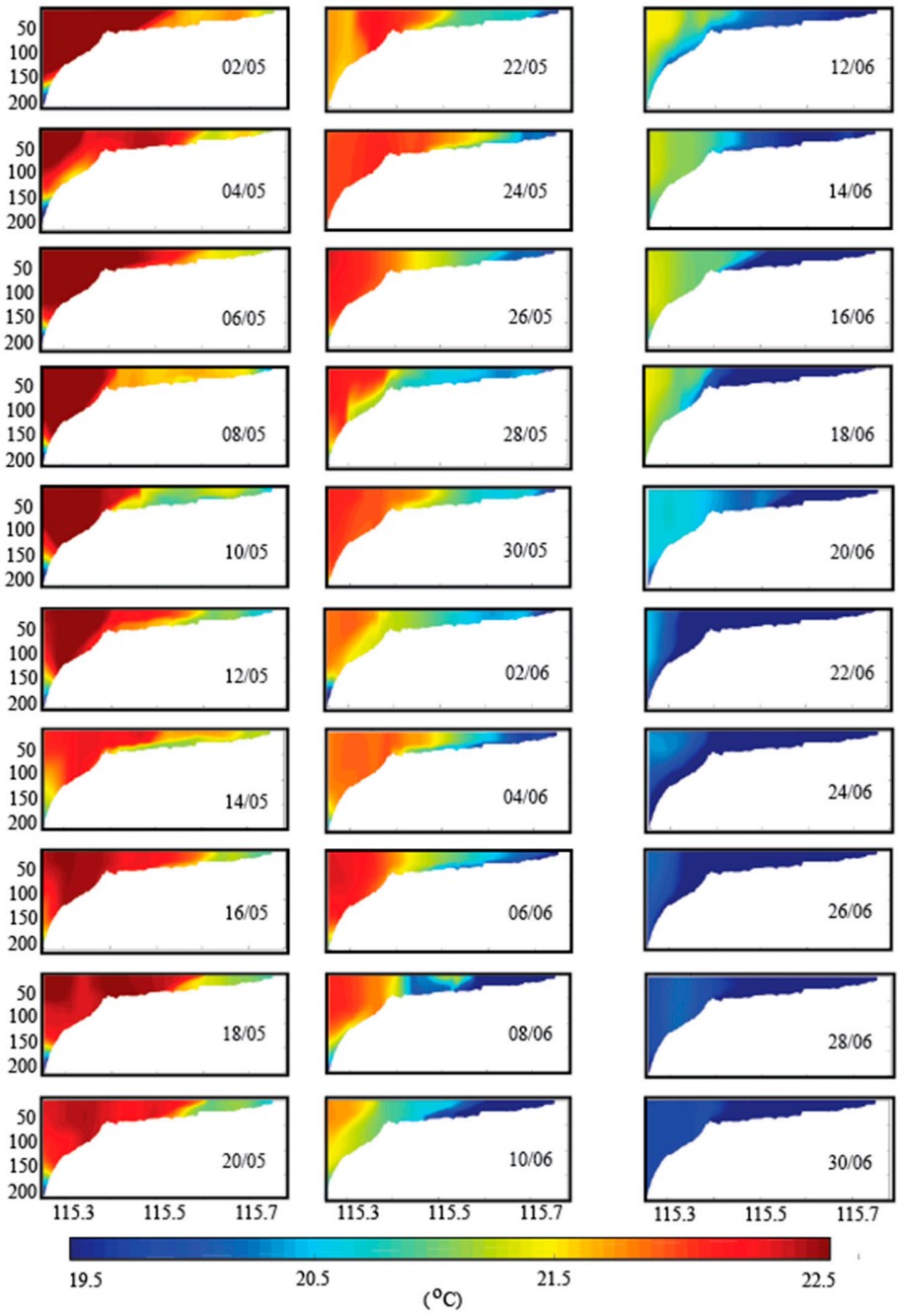

**Figure 6.** Predicted cross-shelf temperature distribution at 2-day intervals for May and June 2016 indicating a cooling trend, intermittent stratification, and formation of DSWC.

### 3.4. Dominant DSWC Flow Patterns

Over the Rottnest Shelf region, several DSWC flow patterns were observed that were associated with specific wind conditions (Figure 2). The DSWC flows were highly variable and sensitive to wind speed and direction and the predicted daily mean bottom velocities were mostly <0.10 ms$^{-1}$ (Figure 7). These flow regimes are illustrated with the distribution of bottom temperature with velocity vectors from the bottom layer of ROMS (Figure 7) and are summarized here with more detail in the subsequent sections. Note that in the following onshore component of winds promote downwelling whilst alongshore wind components promote upwelling.

(I) Weak to moderate NW–SW (May 13–15): onshore component

Southwesterly flow existed over the majority of the domain, driven by southward-directed wind stress, with offshore-directed DSWC flow concentrated to the north of Rottnest Island that diverted the flow offshore to deeper water, including the Perth canyon. This flow pattern is associated with downwelling due to onshore winds. The coldest temperatures originated in Cockburn Sound, exiting along the sea bed to the north and south of Garden Island (Figure 7a).

(II) Weak land/sea breeze conditions with southerlies in the afternoon (May 16–18): alongshore component

Offshore of the continental shelf, the Leeuwin Current flowed southward whilst the shelf waters were forced north by the southerly winds associated with the sea breeze. The DSWC pathways were not as clearly defined despite a band of colder northward-flowing water along the inner shelf and a well-defined gradient between the colder nearshore and warmer offshore waters (Figure 7b). This suggested that DSWC pathways may be more diffuse along the coast under these lower wind conditions, and less concentrated compared to when winds and prevailing currents are from the NW.

(III) Strong winds from the NW due to a cold front (May 19–21): onshore component

Under these conditions bottom currents were directed south or southwestward over the entire domain, similar to regime (I), but with higher velocities and more convergence of offshore-directed flows to the north of Rottnest Island (and onto the Perth canyon) and the shallower water between Rottnest and Garden Islands (Figures 1 and 7c). Colder ocean temperatures near the coast induced by negative heat fluxes associated with the storm were also evident (Figure 7c).

(IV) Moderate to strong winds from the NW due to a cold front (May 22–24): onshore component

This regime was a continuation of (III), with a second storm directing currents toward the southwest, and stronger regions of convergence to the north and south of the islands. However, the inner shelf currents were directed more towards the west (offshore) when compared to event (III). Cooler temperatures extended further offshore, and DSWCs appeared to be concentrated in the same two areas, although it was difficult to determine subtle differences between the two events (Figure 7d).

Strong southerly winds appeared to inhibit DSWC formation due to vertical mixing [2], and these conditions were rare during May 2016, so they are not discussed in detail here with regard to the DSWC pathways. Two contrasting events, a subset of those discussed above including weak winds (May 13–16) and strong onshore NW winds (May 20–23), are analyzed in more detail in the following two sections.

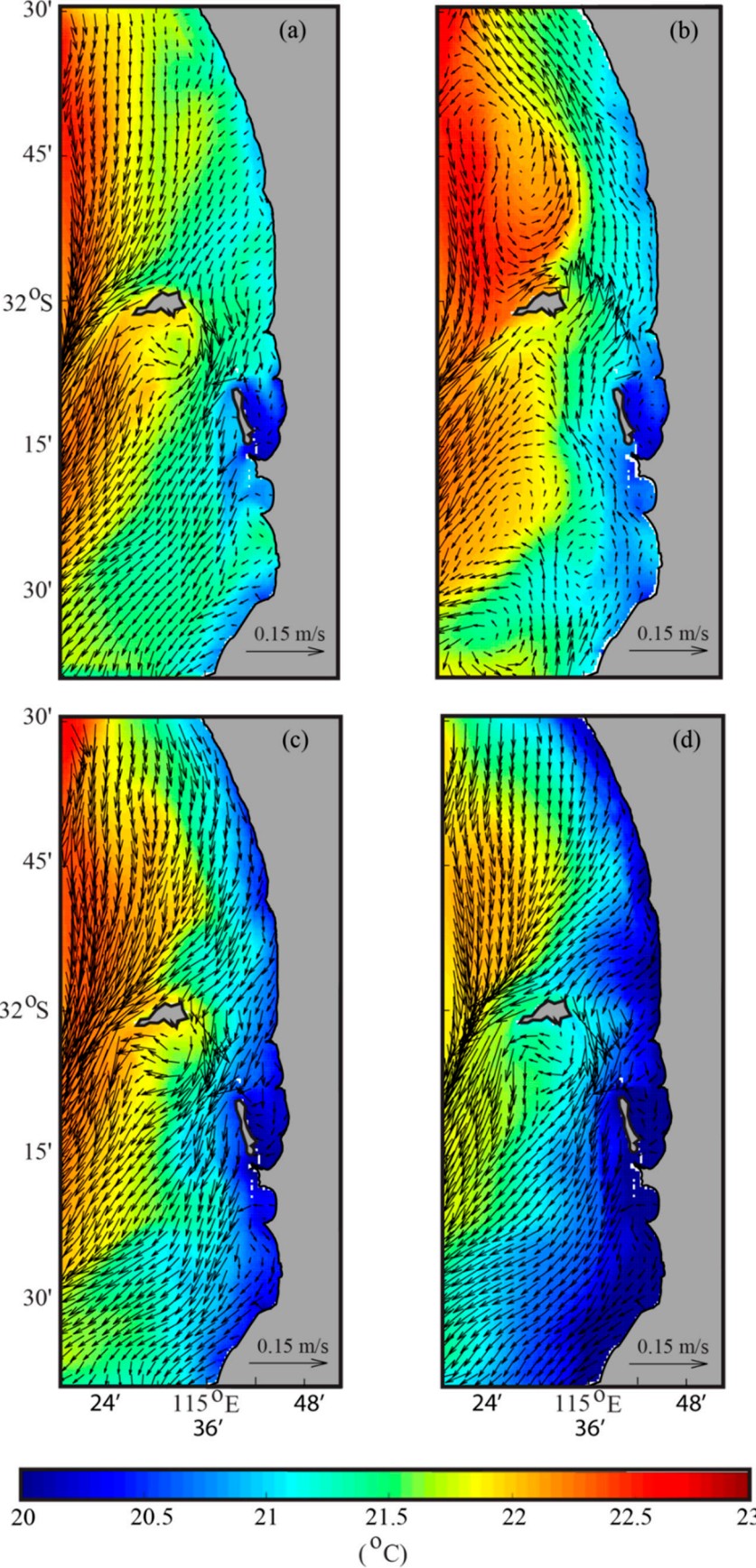

**Figure 7.** Estimated near-bottom velocity and near-bottom temperature 3-day means averaged over four different wind events: (**a**) May 13–15, (**b**) May 16–18, (**c**) May 19–21, and (**d**) May 22–24.

*3.5. DSWC Flow Pathways—Low to Moderate Wind Speeds (Onshore Winds)*

During low to moderate wind speed (<10 ms$^{-1}$) conditions (May 13–16, 2016; Figure 2), up to five DSWC flow pathways (P1–P5 in Figure 8a) could be identified over the model domain using bottom temperature as a proxy. Following five days of wind speeds of <10 ms$^{-1}$ and variable direction (Figure 2), on May 13, the DSWC pathways were evenly distributed along the coast and flowed directly offshore (Figure 8a). As winds started to blow from the northwest late on May 14 and increased from almost zero to ≈10 ms$^{-1}$, the DSWC flows were deflected to the south, and P1 and P2 were less defined and contracted to nearer the coast, whilst P3 and P4 were the dominant flow pathways (Figure 8b). The following day, winds weakened a little but the DSWC pathways were unchanged (Figure 8c). On May 16, with weaker westerly winds, there appeared to be an intermediate state with no clear DSWC pathway in the northern part of the domain and a homogeneous band of cooler water along the coast (except a weak P1). P5 was still clearly defined and appeared to be the main flow pathway under these conditions (Figure 8d).

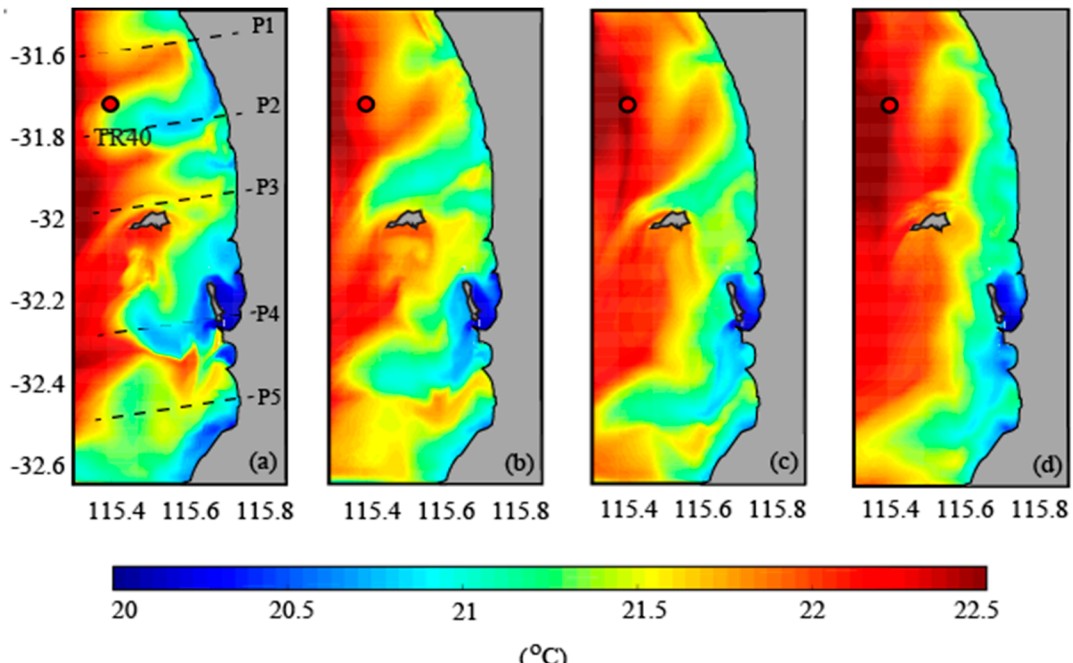

**Figure 8.** Distribution of daily mean bottom temperature showing the spatial and temporal variability of cold water plumes from model output during low to moderate wind conditions (1–10 ms$^{-1}$): (**a**) May 13, 2016; (**b**) May 14, 2016; (**c**) May 15, 2016; and (**d**) May 16, 2016. P1, P2, P3, P4, and P5 were defined as five different DSWC pathways.

In order to examine the vertical structure and location of the DSWC pathways (P1–P5) and assess their spatial and temporal variability, an along-shelf transect was extracted along the 40 m depth contour following the model grid (Figures 1a and 9). In this view, the locations of individual DSWC flowing off the shelf could easily be identified. The DSWC along this depth contour varied in vertical thickness between 10 and 20 m, although P4 and P5 extended over the entire water column. The coldest temperatures were approximately 21 °C. Over the four-day low to moderate winds, the DSWC pathways were deflected southward and converged due to wind conditions and topography (see dashed lines in Figure 9): P2 merged with P3 on May 14 and the combined DSWC pathway remained at that location on subsequent days due to the barrier caused by Rottnest Island to the south (Figure 9b); similarly, P4 was deflected to the south and merged with P5.

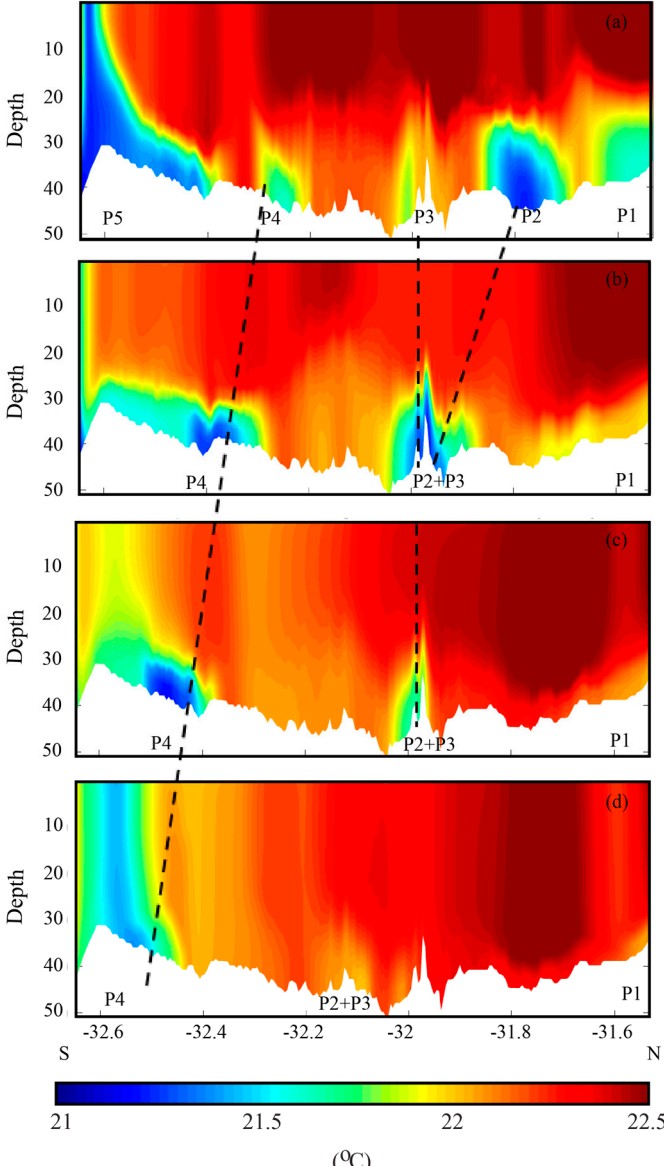

**Figure 9.** Predicted daily mean temperature along a shore-parallel transect (view from land with south located to the left) along the 40 m depth contour: (**a**) May 13, 2016; (**b**) May 14, 2016; (**c**) May 15, 2016; and (**d**) May 16, 2016. P1–P5 are related to the flow pathways defined in Figure 8.

Due to the deflection and merging of DSWC pathways under low to moderate wind conditions, describing their occurrence based on fixed cross-shelf transects and bottom temperature only provided a somewhat misleading view as the water column information was missing (Figure 10). On May 13, following several days of weak winds, all four cross-shelf transects indicated well-developed DSWCs with vertical temperature stratification on the shelf (Figure 10a–d). This contrasts with Figure 8 where only the offshore extent of the DSWCs could be identified. Under these conditions, a single transect within the study domain would have been sufficient to identify the presence of DSWC. This is consistent with Figure 8 and the discussion above. However, on subsequent days, under the influence of northerly winds, the DSWC pathways migrated southward and the northerly transects (P1 and P2) appeared to be vertically mixed and the DSWCs shut down. At the same time, the more southerly transects showed persistent stratification and DSWCs reaching near the continental shelf edge (Figure 10). By May 16, following a wind directional change to the south, all transects were again similar, with a nearly vertically mixed water column and minimal DSWCs (Figure 10).

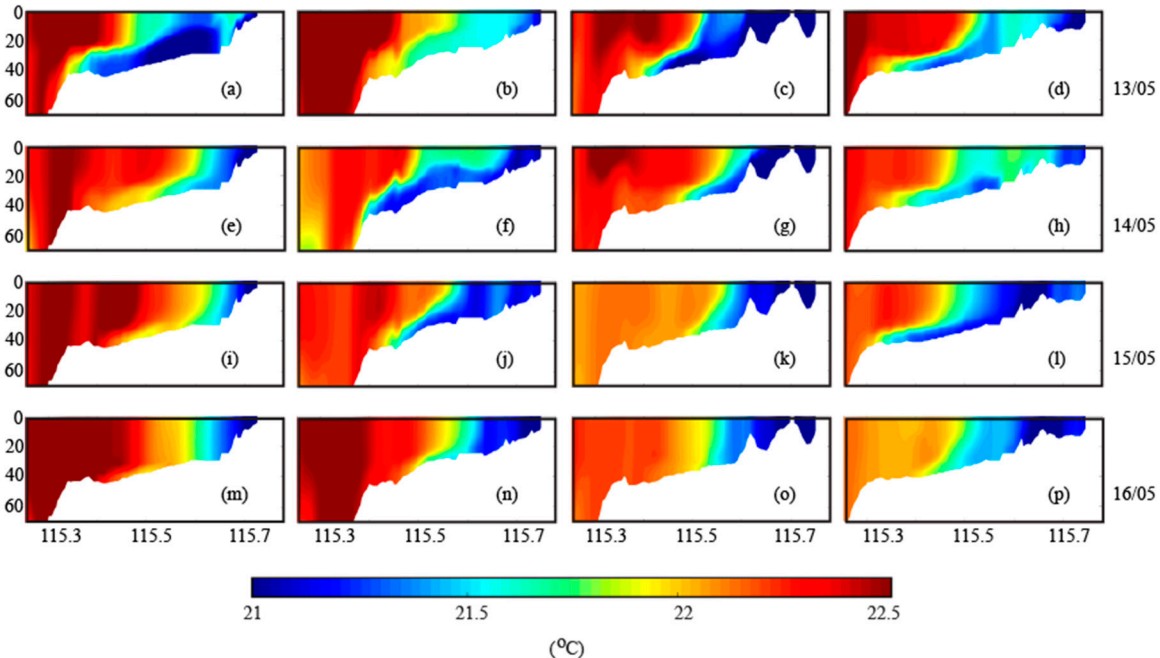

**Figure 10.** Cross-shore transects during low to moderate wind conditions (May 13–16, 2016) for four different latitudinal locations: (**a,e,i,m**) −31.8° N (P2); (**b,f,j,n**) −31.95° N (P3); (**c,g,k,o**) −32.25° N (P4); and (**d,h,l,p**) −32.44° N (P5). P1–P5 are related to flow pathways defined in Figure 8.

Predicted near-bed velocities confirmed the scenarios described above, highlighting the difference between calm, northerly, or southerly wind conditions. Weak southerly winds resulted in northward flow along the coast with dense water from Cockburn Sound flowing off of the shelf to the north of Rottnest Island (Figure 11a). During weak northeasterly winds, bottom velocities indicated that DSWCs were deflected to the south with maximum velocities on the order of 0.05–0.10 ms$^{-1}$, although through this basic analysis, it was not possible to separate the baroclinic component from the barotropic (Figure 11b). Without southerly winds, it appeared that the cooler water in Cockburn Sound flowed offshore around Garden Island (Figure 11b).

*3.6. DSWC Flow Pathways—Strong Winds (Onshore)*

Following the weak wind conditions described in Section 3.5, the region experienced a storm event with intense winds (maxima > 20 ms$^{-1}$) blowing from the north and west (i.e., with an onshore component) over three days (Figure 2). The response of the shelf waters and DSWC were similar to the description given in the previous section, but with stronger currents, more vertically mixed conditions, and stronger horizontal temperature (density) gradients. Intense cooling of inshore waters due to the passage of cold front was clearly evident over the four-day period (Figure 12).

Prior to the arrival of the cold front on May 20, all five of the DSWC pathways (P1–P5) were again visible and flowed directly offshore (Figure 12a). Over the next two days, P2 intensified and merged with P1, and was deflected southward to Rottnest Island by the northwesterly winds (Figure 12b,c). South of Rottnest Island, it appeared that the cold dense water was mostly advected southward, parallel to the coast, whilst winds were strong and from the north (P3–P5; Figure 12b,c). When winds began to weaken and turned more westerly, P4 and P5 were once again directed offshore (Figure 12d).

The along-shelf 40 m depth contour temperature section (Figures 1a and 13) showed vertically mixed conditions except at P2 and P3, where strong vertical stratification was evident over the final two days of the storm. At P2, north of Rottnest, the cold bottom layer was >20 m thick (Figure 13c). P4 and P5 were combined with no vertical stratification present and the coldest temperatures over the entire domain (<21 °C) (Figure 13d).

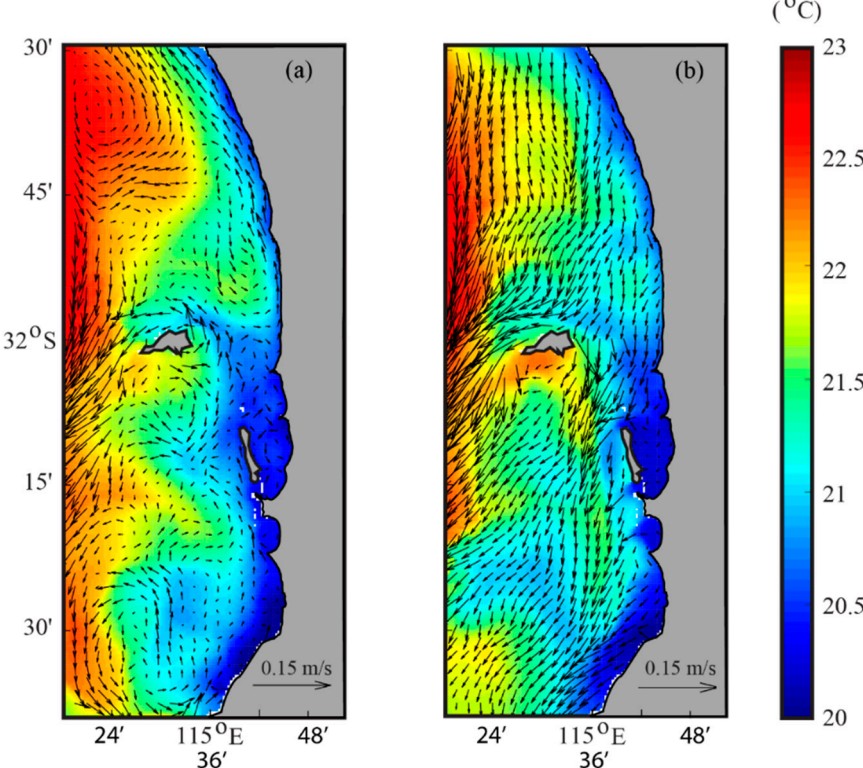

**Figure 11.** Predicted near-bottom mean velocity and near-bottom temperature during two low wind events: (**a**) May 7–9, 2016; and (**b**) May 10–12, 2016. The near-bottom velocities and temperature were averaged over 3 days.

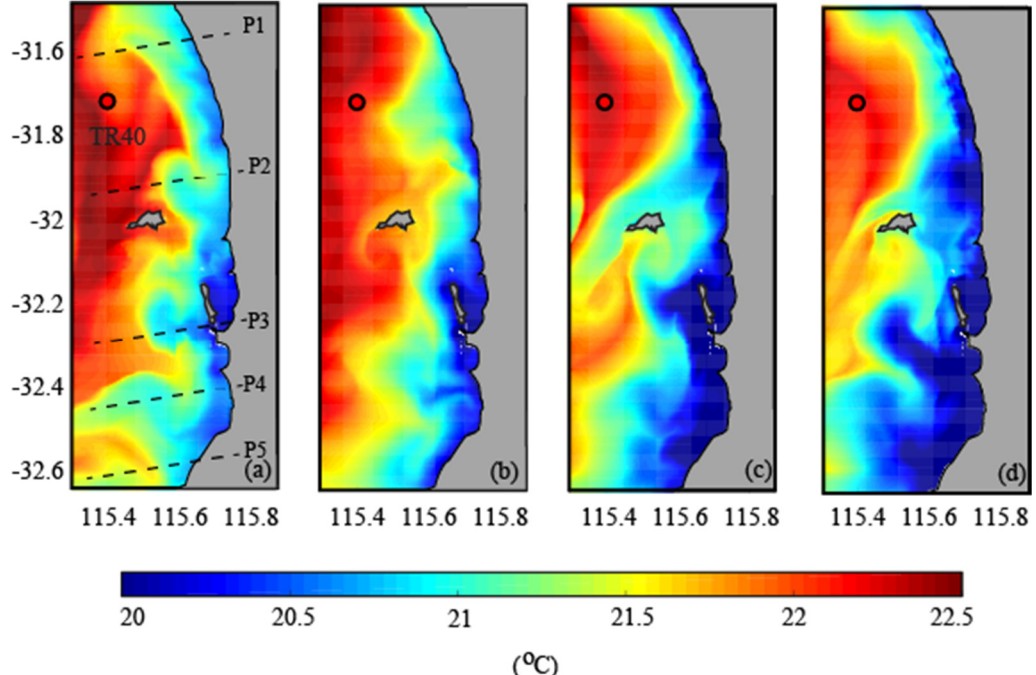

**Figure 12.** As for Figure 8 but during strong wind conditions (20 ms$^{-1}$): (**a**) May 20, 2016; (**b**) May 21, 2016; (**c**) May 22, 2016; and (**d**) May 23, 2016.

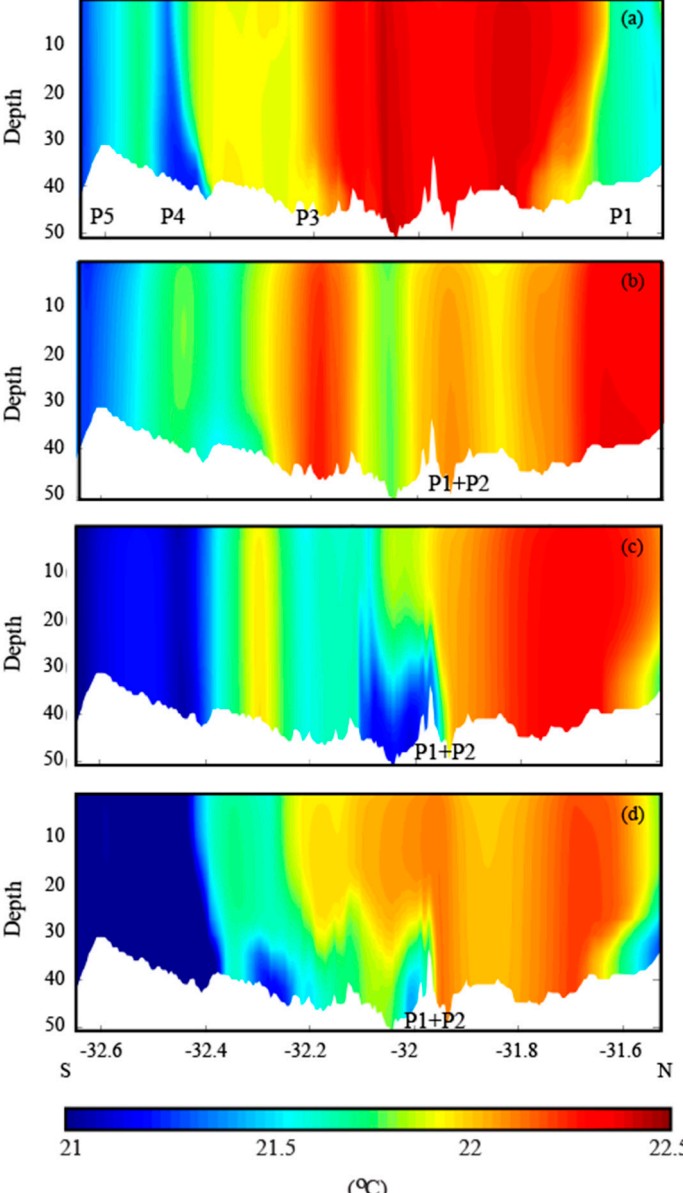

**Figure 13.** As for Figure 9 but during strong wind conditions (20 ms$^{-1}$): (**a**) May 20, 2016; (**b**) May 21, 2016; (**c**) May 22, 2016; and (**d**) May 23, 2016.

The cross-shore temperature transects showed evidence of strong vertical mixing, downwelling enhancement of DSWCs, cooling, and intensification of the horizontal temperature (density) gradient (Figure 14). Although the water column was vertically mixed, a horizontal density gradient was always present. Similar to the low to moderate wind events, the location of the transect was critical in order to accurately describe the existence of the DSWCs, with more cascades in northern transects appearing to cease whilst the more southerly transects (P3,P4) maintained DSWC characteristics (Figure 14j,k,n,o,p).

The fact that P3 intensified in the presence of strong winds (>20 ms$^{-1}$) and maintained vertical stratification suggested that downwelling due to the strong onshore (northwest) winds acted to intensify the DSWCs. The rapid heat loss during and after the storm would have also acted to enforce DSWCs, although the strengthening effects of the density gradient would have probably been secondary compared to the effect of downwelling until the vertical wind mixing reduced.

During this period, strong onshore (northwest) winds generated southward transport over the entire shelf was a factor of 10 greater than the cross-shelf transport. This suggested that, under these conditions,

dense inshore waters over the entire region may have been advected large distances parallel to the coast before being forced offshore as downwelling-enhanced DSWCs at locations where topography aids in directing the flows off the shelf (e.g. P2 north of Rottnest Island; Figures 12c and 13c).

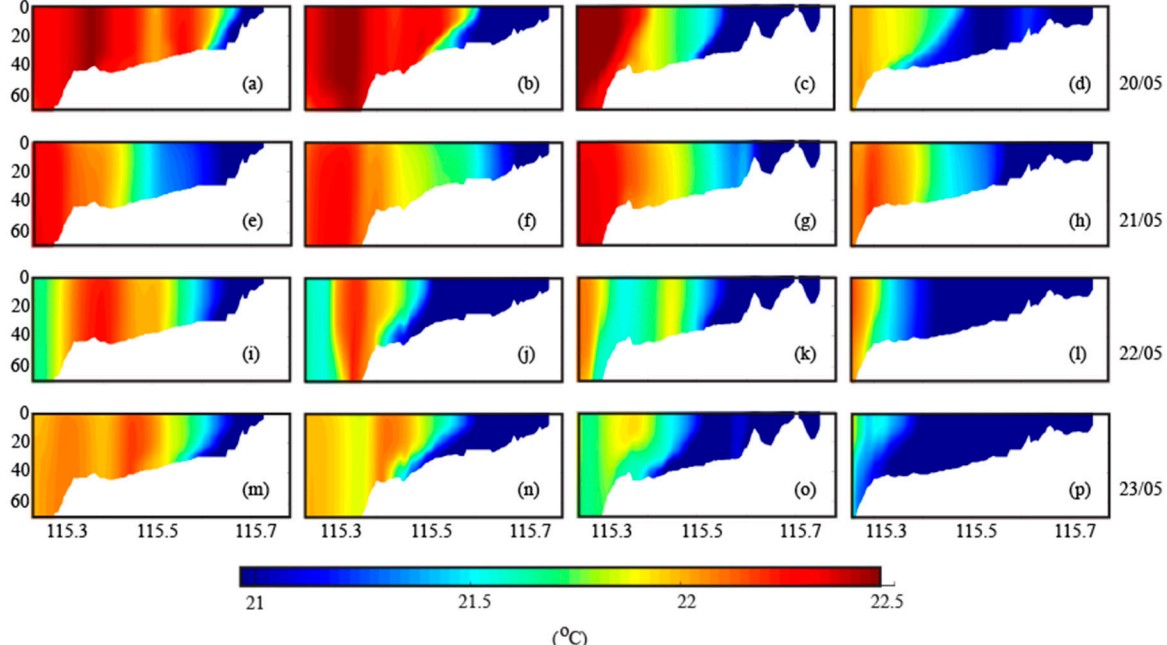

**Figure 14.** As for Figure 10 but during strong wind conditions (20 ms$^{-1}$): (**a,e,i,m**) −31.8° N (P2); (**b,f,j,n**) −31.95° N (P3); (**c,g,k,o**) −32.25° N (P4); and (**d,h,l,p**) −32.44° N (P5). P1–P5 are related to flow pathways defined in Figure 12.

### 3.7. Ocean Glider Data: May 13–26, 2016

An ocean glider was deployed along the Two Rocks region over the period May 13–26, 2016, and obtained six cross-shelf transects that included the changing wind conditions (Figure 15). One of the glider transects was used to validate the ROMS model (Figure 4). For the first transect (May 13–15) the prevailing conditions were moderate onshore winds up to 10 ms$^{-1}$ that were preceded with low wind conditions (Figure 2). Here, a well-established DSWCs was present with both temperature and salinity determining the cascade (Figure 15a). The next transect (May 15–16) was similar with onshore winds; the data were almost the same except that the cascade extended to deeper >150 m water (Figure 15b). The period May 17–18 represented low wind conditions with a DSWCs present on the shelf (Figure 15c). Over the period May 18–20, the wind speeds were relatively low but the direction changed to southerly and the signature of the cascade was diminished (Figure 15d). This could be due to northward currents moving the cascade to the north out of the ocean glider transect (Figure 7b). The peak of the first storm was at midnight on May 2 (Figure 2). The next ocean glider transect was May 22–23 and included the period after the peak of major storm and also the peak of the second storm. The impact of the two storm systems were reflected with the transition to well-mixed waters (both salinity and temperature) on the inner shelf. There was also cooling of the entire transect with upper shelf cooling by >1 °C and offshore regions by >0.6 °C (Figure 15e). Similarly, there were changes in salinity with the upper shelf increasing in salinity by 0.1 through vertical mixing. The subsequent transect (May 23–25) had decreasing wind speeds with changing direction and the initiation of a cascade was evident in the temperature record (Figure 15f). This sequence of changing conditions were reflected in the model output (Figure 14).

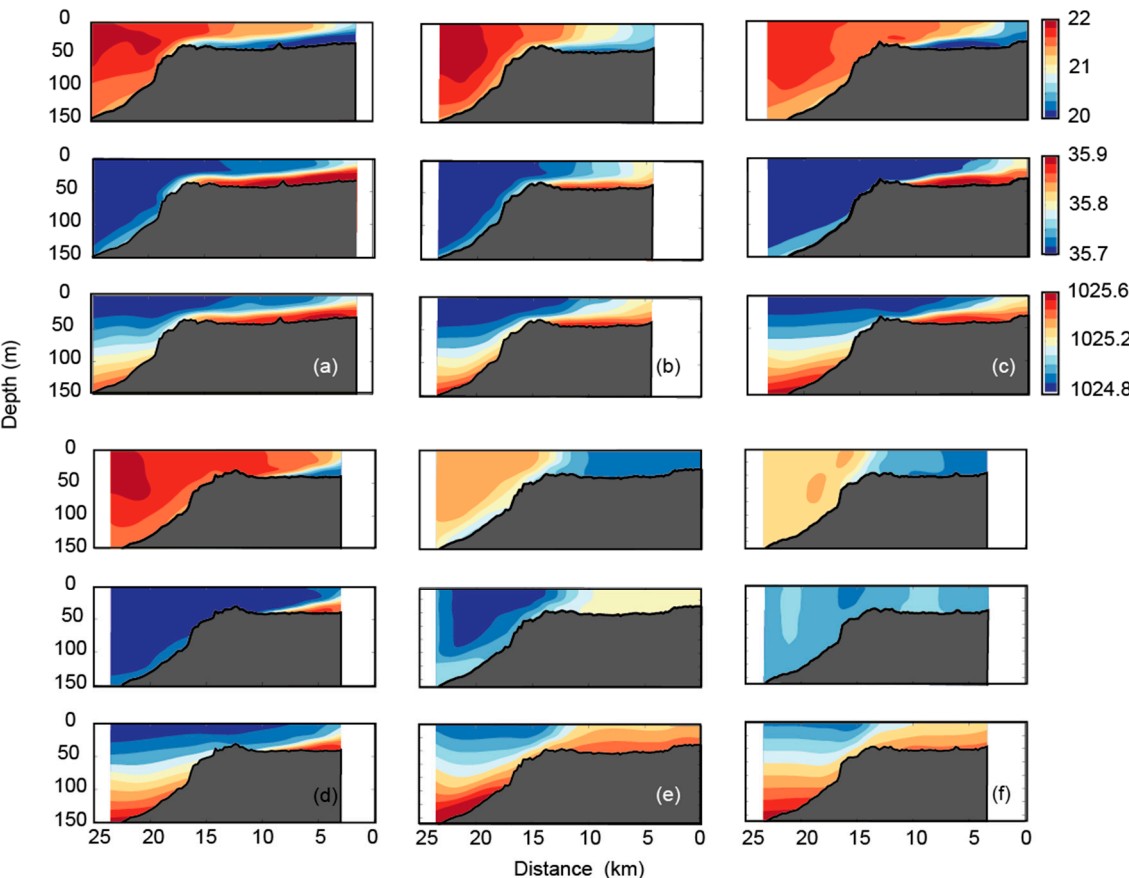

**Figure 15.** Temperature (°C), salinity (psu), and density (kg/m$^3$) along the Two Rocks transect collected between May 13–26, 2016, using an ocean glider: (**a**) 0912 May 13–0300 May 15; (**b**) 0330 May 15–1600 May 16; (**c**) 2240 May 16–1200 May 18; (**d**) 1240 May 18–0200 May 20; (**e**) 0720 May 22–2320 May 23; (**f**) 2345 May 23–2345 May 25.

## 4. Discussion

In this paper, the circulation model ROMS with real-forcing was used to examine the spatial three-dimensional structure and temporal variability of DSWCs along the Rottnest continental shelf (RCS). In Mahjabin [2], features of DSWCs around Australia was examined using ocean glider data and included large spatial scales and seasonal (monthly) variability. Data from moorings and ocean gliders were used to examine DSWC formation along a single transect (Two Rocks) over a period of ≈1 month [2]. DSWC formation along the RCS were also studied for a year in a single location where ocean glider data were available [5]. Both of these studies were limited to two dimensions (cross-shelf transect). It was found that the cross-shelf density gradient was the dominant forcing mechanism for the formation of DSWC. Both wind speed and direction had a strong influence that either inhibited (through vertical mixing associated with offshore or southerly winds) or enhanced (through downwelling associated with onshore winds) the DSWCs. The focus of this paper is on the three-dimensional aspects, and in particular, the definition of flow pathways, along-shelf movement of DSWCs, and response to passing weather systems (cooling and mixing events and changes in wind direction).

This study is one of few numerical modelling studies that addresses the behavior of DSWC on a continental shelf. Overall, studies that have addressed DSWCs in subtropical regions where surface cooling is the major driving mechanism are rare (in polar regions salinity changes through ice formation is the dominant mechanism). These have concentrated on extreme events that contributed to DSWCs during the winter of 2012–2013 in the Adriatic Sea [46] and in the Gulf of Lions [52], and included large

model domains (>1000 km). This paper examined processes at a smaller scale (≈200 km) resolving finer scale features. A recent study by Wu et al. [23] examined the effects of surface cooling along the mid-Atlantic Bight in the formation of DSWCs but was limited to a 2-D cross-sectional transect.

Surface heat flux generally changes on two major time scales: day/night and the seasonal cycle. The influence of the seasonal cycle was demonstrated in Mahjabin 2018 [2] where the majority of DSWCs occurred during the winter months due to differential heat loss generating a cross-shelf density gradient. This paper demonstrated that, in addition to the daily cycle, rapid heat loss (to a max of 800 Wm$^{-2}$ and a mean of 420 Wm$^{-2}$ over 24 h) occurred during the passage of a cold front associated with an extratropical storm system (Figure 2). This cooling rate, over the 24 h period, is equivalent to a decrease in depth mean temperatures by ≈1 °C and 0.3 °C in 10 and 30 m water depths, respectively [2]. Thus, the passage of associated cold fronts can enhance the horizontal temperature/density gradients.

During May 2016, the main period of this study, a range of wind conditions was experienced in the study region due to the west to east passage of extratropical low/high pressure systems (Figure 2). Three frontal systems made land fall with maximum wind speeds >15 ms$^{-1}$. During the intervening periods between the storms, the winds were generally <10 ms$^{-1}$ with variable directions (Figure 2). Under lower wind conditions, five different DSWC pathways were identified using bottom temperature as a proxy (Figures 8 and 10). As the wind speed increased from the northwest, the DSWCs were deflected southward and some of the flows merged when they were controlled by topography. In particular, the presence of Rottnest Island was a major influence. When a strong onshore wind prevailed (associated with passage of cold fronts), the DSWC were deflected strongly to the south and was enhanced through downwelling: strong vertical stratification and offshore flow being present along the main pathway regions (Figures 7 and 14). The most consistent DSWC flow pathways were identified to be located to the northern side of Rottnest Island, and south of Garden Island, with bottom topography playing a significant role. Here, the presence of Rottnest Island deflected the flow offshore to deeper water and the head of Perth canyon. Studies in the Gulf of Lions [53,54] indicated that the DSWCs can extend to depths to 1000 m. However, absence of data (and the model domain) did not allow us to examine the extent of the DSWCs in the canyon. The ocean glider indicated that DSWC along the Two Rocks transect extended to depths >150 m [2].

Previous studies on DSWC formation along the RCS were based on cross-shelf transects in a single location where ocean glider and/or mooring data were available [2,5]. Numerical modelling results indicated although the Two Rocks transect (Figure 1) was located along one of the flow pathways (P1; Figure 8), the main outflows occurred to the south (northern side of Rottnest Island, and south of Garden Island). Numerical model results and satellite derived sea surface temperature imagery indicated cooling of the nearshore region along the whole coastline (Figures 3 and 7). This would generate a cross-shelf density gradient and drive DSWCs, which was confirmed by the numerical model. Thus, the observations along Two Rocks are representative of the processes along the entire study region although the model output indicated more consistent DSWCs to the south.

Cross-shelf transport on the continental shelf is major feature of the coastal circulation [27,55,56]. Examining processes along a single cross-shelf transect enables examination of the important processes; however, three-dimensional circulation patterns from model output indicated that the longshore processes were also important. In particular, the prevailing currents, in this case driven by winds, deflected the currents (from the cross-shelf transect) either north or south depending on the incident wind direction.

Numerical model results also confirmed that both wind speed and direction were important in the formation of the DSWC cascades. The passage of a storm front, in the local study region, resulted in intense cooling of the coastal zone accompanied by strong onshore winds (winds changing from northwest to west to southwest). Both the numerical model and ocean glider data obtained during May 2016 indicated that the water column responded almost instantaneously to surface cooling and the changing wind conditions (Figures 12, 14 and 15). The moderate onshore winds caused downwelling that resulted in the extension of the colder water further offshore along the sea bed (Figure 12). Strong

winds also completely mixed the water column and when the wind speeds decreased, the cascade was re-established (Figures 14 and 15).

The application of a numerical circulation model provided a valuable tool that improved the understanding of DSWC due to strong cooling by atmospheric cold fronts that generated cross-shelf density gradients, and highlighted the three-dimensional aspects of the cascades; in particular, the presence of preferential pathways and the spatial distribution of DSWCs in the alongshore. The results indicated that cooling of the nearshore region will allow for the generation of DSWCs along the coast. This result can be extended to the Australian coastline where the climatology of sea surface temperatures indicated [2] many regions that indicate cooler water at the coast that most likely have DSWCs. Hence, findings are directly transferrable to other regions around Australia and globally as the processes described here may occur in many continental shelf seas, particularly in the sub-tropical regions.

The numerical model results, combined with the glider observations around Australia [2], provide further evidence that DSWCs are an important component of cross-shelf transport from shallow inshore waters into the deep ocean. The broad occurrence of DSWCs around the Australian continental shelves (at scales of tens to thousands of kilometers) suggested that these density-driven processes are important for cross-shelf as well as alongshore movement of any suspended matter including larvae, nutrients, salt, heat, carbon, low-oxygen water, sediment, and pollutants.

## 5. Conclusions

Numerical simulations using a validated ocean circulation model were used to examine the spatial and temporal variability of dense shelf water cascades (DSWCs) in a wind-dominated region (Rottnest Continental Shelf) where DSWC are a common occurrence during autumn/winter months.

Results indicated large spatial and temporal variability of DSWCs, and the strengthening of the cross-shelf density gradient due to seasonal and short-term cooling events. Several dominant flow pathways were defined, which responded to changes in the mean wind conditions, with topography playing an important role. Wind intensity and direction strongly influenced DSWC formation and evolution with persistent DSWCs observed during low wind conditions, and stronger winds modifying flow pathways. The passage of atmospheric cold fronts associated with winter storms resulted in rapid heat loss through evaporative cooling. These conditions resulted in enhancement of the DSWCs due to modifications of the cross-shelf density gradient and wind effects. Specifically, onshore (offshore) directed winds resulted in an enhancement (inhibition) of DSWC due to downwelling (mixing). Consequently, the largest DSWC events occurred during the passage of cold fronts when atmospheric temperatures reinforced favorable horizontal density gradients and onshore winds promoted downwelling that enhanced DSWCs. Advection of dense shelf waters was also strongly influenced by the wind conditions, with significantly more transport occurring along-shelf compared to cross-shelf under most conditions; this aspect needs to be considered when interpreting field data. The broad occurrence of DSWCs on the Australian continental shelf (at scales of tens to thousands of kilometers) suggested that these density-driven processes are important for cross-shelf as well as alongshore movement of any suspended matter including larvae, nutrients, salt, heat, carbon, low-oxygen water, sediment, and pollutants.

**Author Contributions:** This study was done as a part of PhD research by T.M. All model data analysis and validation were done by T.M. with the supervision of C.P. and Y.H. The ROMS model output used for this study was generated by I.J.

**Funding:** This postgraduate research was funded by Scholarship International Research Fees (SIRF), University International Stipend, University Postgraduate Award, and Ad Hoc Postgraduate Scholarship. The collection of oceanographic data was funded through Integrated Marine Observing System (IMOS), a national collaborative research infrastructure program, supported by the Australian Government.

**Acknowledgments:** The ocean glider and oceanographic mooring data used in this paper were sourced from Australia's Integrated Marine Observing System (IMOS). The ocean glider data were collected by the IMOS Ocean Glider facility located at the University of Western Australia. The mooring data used in this paper were collected

by the Commonwealth Science and Industrial Research Organisation (CSIRO), a sub-facility of the Australian National Mooring Network (ANMN). The meteorological data were obtained by the Bureau of Meteorology. The authors acknowledge the support of Dennis Stanley, Paul Thomson, Kah Kiat Hong, and Mun Woo for the tasks associated with the deployment and recovery of the gliders, glider piloting, and processing the data Quality Assurance Quality Control (QA/QC) of the ocean gliders. All IMOS data (SST, ocean glider, and moorings) are available through the Australian Ocean Data Network (https://portal.aodn.org.au).

**Conflicts of Interest:** The authors declare that the research was conducted in the absence of any commercial or financial relationships that could be construed as a potential conflict of interest.

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
