# Peer review of "Spatial and Temporal Variability of Dense Shelf Water Cascades along the Rottnest Continental Shelf in Southwest Australia"

_jmse, doi:10.3390/jmse7020030_

Round 1

Reviewer 1 Report

Spatial and Temporal variability of Dense Shelf Water Cascades along

the Rottnest continental shelf in south-west Australia

by Tanziha Mahjabin, Charitha Pattiaratchi, Yasha Hetzel, and Ivica

Jankovic

--- Summary ---

Mahjabin et al present a descriptive analysis of a ROMS simulation of

the Western Australia continental shelf near Perth, focusing on the

early-winter formation of Dense Shelf Water Cascades (DSWC) by surface

buoyancy loss and dense water formation near the coast. The

description includes cross-shelf sections and plan-view maps of bottom

water characteristics and their temporal evolution in response to

atmospheric forcing events, including low-wind periods and storms

during May and June 2016.

--- General Comments ---

The detailed description of the ROMS model results does a good job of

presenting the complex spatial structure of the dense plumes (both in

plan view and in progressive cross-shelf sections) and their temporal

variability.  Also, the presence of similar cross-shelf structures in

the dense bottom water in the glider sections lends support to the

idea that similar DSWC processes are in fact occurring in the real

world.

In fact, the overall presentation is quite complete, and publication

could conceivably be permitted after fairly minor revisions. However,

I have several suggestions and questions which, if addressed, I think

would greatly improve the impact and relevance of the work to the

community. If not in the current paper, I hope the authors will

consider the following in future work:

1) The manuscript at present is almost entirely qualitative, and

provides no estimates of DSWC volume, areal extent, export flux,

variability, or dynamical scaling relating these quantities to wind,

buoyancy forcing, or baroclinic instability--all of which can

certainly be obtained with the model output, and to some degree also

with the observations. This is very unusual for scientific papers,

especially considering that the dense water flows are not exactly a

new phenomenon. The temporal evolution and spatial structure are shown

in considerable and fascinating detail, but this would be much more

valuable if accompanied by at least a few quantitative results.

2) The background information presented is fairly limited and does not

include reference to many of the past analytical, laboratory, and

numerical studies of the transport and instability of dense gravity

currents--either from localized sources or from broad regions of

continental shelves such as in the current situation. Good starting

points for the former include Jiang and Garwood (JPO 1996) and

Lane-Serff and Baines (JFM 1998), and for the latter Gawarkiewicz and

Chapman (JGR 1995) and subsequent papers by David Chapman in JPO and

JGR, as well as Pringle (JGR 2001). Many of these papers lay out

scaling arguments connecting dense water flux with parameters such as

slope, shelf width, and buoyancy forcing, which could be compared and

calibrated with the observational and numerical model results in the

current manuscript. This could be a good way to get DSWC predictive

capability from the limited glider observations or simply from the

surface fluxes.

3) The dynamical descriptions that are presented are not always

accurate, and seem to ignore the effect of the Coriolis force,

inertial oscillations, and Ekman transport in determining the wind

forcing of the coastal circulation (l.21,l.309-310,l.488-489,

l.578-580). For example, it should be the southward, along-shore,

component of the wind which sets up the on-shore Ekman transport that

drives downwelling (and not the on-shore wind component). In addition,

the claims about "cascading" dense water (e.g., l.296-297,Fig.15) are

not always verified by the cross-shelf snapshot sections shown, since

the horizontal density gradients can be in geostrophic balance with a

vertical shear. Again, this balance can be evaluated with the model.

4) Validation of the model really should be done using some sort of

quantitative metrics, rather than simple side-by-side descriptive

figures (though these are certainly informative). Some options include

the integrated dense water volume or density anomaly in sections, the

transport of geostrophic along-shelf currents (also in sections), or

patchiness statistics of the dense water thickness or reduced gravity.

5) The reasoning behind the choice of the numbered dense water

pathways (Fig.8) is not clearly laid out, and the fact that they often

move and merge over time (l.369-373) calls into question the utility

of this labeling. Although some of the pathways do make sense to label

because they coincide with topographic channels or paths around

barriers (islands), the pathways over the broad slope seem better

described by an average spacing determined by fluid dynamic scaling

arguments (such as rossby radius or baroclinically unstable mode).

6) The direct sampling of the numerical model at the transit speed of

the glider (Fig.4) suggests that the model could be used as an

extrapolation tool, providing a means to relate the actual

instantaneous snapshot sections to the slow and undersampled glider

sections.

7) Finally, a most basic question about the DSWC process that does not

seem to be answered by the paper is: What is responsible for the

generation of dense water at the coast? Some possible explanations

include (a) spatial gradients in surface heat and freshwater fluxes,

(b) a gradient in the depth over which the surface fluxes operate

(e.g., shallowing near the coast along with the bathymetry), (c)

upwelling of cold water at the coast preconditioning the shelf for

broad cooling/evaporation and subsequent relaxation and coastal DSWC

or (d) shelf circulation leading to retention of waters through the

cooling season near the coast where they can be acted upon by surface

forces for a longer time. Since the ROMS model does reproduce the

spatial structure of the dense water fairly well, it should be able to

provide the answer. But without spatial maps of surface flux,

mixed-layer depth, or residence time, currently the reader can only

guess.

--- Minor Comments/Typos ---

l.217: M1 and M2 locations are not shown in Fig.1, and there is no

Fig.1a labeled. Is this figure missing?

l.219: Cumulative net heat loss should have units of J m^-2 instead of

W m^-2.

Fig.2: The wind direction (panel b) often has steep downward slopes

indicating counterclockwise rotation in approximate resonance with

the inertial period. Is there much evidence of inertial oscillations

in the simulations following these events?

Fig.5: Caption and figure labels are missing a few items, including

whether panels c,d,g, and h are ADCP or model, U or V, and averaged

over event A or B. I'm guessing panel c is ADCP U for event A, d is

ADCP U for event B, and g and h are ADCP V for the two events.

l.383-384: Given the spatial inhomogeneity and large gaps between

plumes shown in Figs.8 and 9, it seems odd to suggest that a single transect

would be sufficient to identify DSWC.

Author Response

Response to reviewers: JMSE 422121 (Submitted to Journal Marine Science and Engineering)

Spatial and Temporal variability of Dense Shelf Water Cascades along the Rottnest continental shelf in south-west Australia

by Tanziha Mahjabin, Charitha Pattiaratchi, Yasha Hetzel and Ivica Janekovic.

We would like to thank and acknowledge the three reviewers and the editor for their careful reading and constructive comments on the manuscript. We believe that we have addressed the issues raised by reviewers and the proposed changes to the manuscript are detailed in this document. We trust that the reviewers and the editor will find that the suggested changes will make the manuscript to be suitable for publication.

In the following, black indicates the comments by the reviewer; blue is our response to the reviewers.

Response to Reviewer #1

The detailed description of the ROMS model results does a good job of presenting the complex spatial structure of the dense plumes (both in plan view and in progressive cross-shelf sections) and their temporal variability. Also, the presence of similar cross-shelf structures in the dense bottom water in the glider sections lends support to the idea that similar DSWC processes are in fact occurring in the real world.

In fact, the overall presentation is quite complete, and publication could conceivably be permitted after fairly minor revisions. However, I have several suggestions and questions which, if addressed, I think would greatly improve the impact and relevance of the work to the community. If not in the current paper, I hope the authors will consider the following in future work:

We acknowledge that the paper entirely qualitative to a certain extent that was the aim as the paper is an extension of previous papers. Of course application of a one-way coupled atmospheric ocean circulation model is not a trivial exercise. The main aim of the model to examine the wider occurrence of DSWC across the shelf and how they respond to variable atmospheric forcing (wind speed and direction, heat fluxes). As it is the paper is quite long (24 pages including 15 Figures).

We appreciate the suggestions of the Reviewer#1 in terms of future and definitely consider the suggestions

3) The dynamical descriptions that are presented are not always accurate, and seem to ignore the effect of the Coriolis force, inertial oscillations, and Ekman transport in determining the wind forcing of the coastal circulation (l.21,l.309-310,l.488-489, l.578-580). For example, it should be the southward, along-shore, component of the wind which sets up the on-shore Ekman transport that drives downwelling (and not the on-shore wind component). In addition, the claims about "cascading" dense water (e.g., l.296-297,Fig.15) are not always verified by the cross-shelf snapshot sections shown, since the horizontal density gradients can be in geostrophic balance with a vertical shear. Again, this balance can be evaluated with the model.

The role of Coriolis force and Ekman transport is an interesting subject. In the upper ocean the transport is confined to a surface boundary layer with a thickness characterized by the Ekman depth δE=√((2Nz)⁄f) assuming a constant eddy viscosity Nz with f being the Coriolis frequency. In water depths < δE transport is entirely in the direction of the wind stress with minimal influence of the Coriolis force (Lenz and Fewings, 2012). In the study region δE~70m (Mihanovic et al., 2016) and thus the inner continental shelf can be considered to be in a nonrotating frame. Under these conditions, when the wind blows towards (away from) the coast, water near the surface moves onshore (offshore). Under onshore wind conditions, a pressure gradient is set-up to balance the wind stress, and that pressure gradient drives an offshore return flow in the lower portion of the water column, similar to a DSWC. Under offshore winds, the surface layer moves offshore and onshore flow is present along the sea bed. Lenz and Fewings (2012) showed that for onshore winds blowing over a vertically stratified shelf (1) onshore flow in the surface boundary layer moving to shallower water combined with vertical mixing forms a vertically mixed inner-shelf region; and, (2) in the stratified region on the middle and outer shelves, there is offshore flow in the bottom boundary layer. These features were observed in the glider transects during events B and C where a vertically mixed region was present at the shoreward end of the transect and stratified conditions on the shelf.

7) Finally, a most basic question about the DSWC process that does not seem to be answered by the paper is: What is responsible for the generation of dense water at the coast? Some possible explanations include (a) spatial gradients in surface heat and freshwater fluxes, (b) a gradient in the depth over which the surface fluxes operate (e.g., shallowing near the coast along with the bathymetry), (c) upwelling of cold water at the coast preconditioning the shelf for broad cooling/evaporation and subsequent relaxation and coastal DSWC or (d) shelf circulation leading to retention of waters through the cooling season near the coast where they can be acted upon by surface forces for a longer time. Since the ROMS model does reproduce the spatial structure of the dense water fairly well, it should be able to provide the answer. But without spatial maps of surface flux, mixed-layer depth, or residence time, currently the reader can only guess.

Previous papers have shown that it is due to differential heating and cooling – your explanation (b). This is a feature that occurs around whole of Australia. Thank you for the suggestion and we will include these features in future work.

Minor Comments/Typo

l.217: M1 and M2 locations are not shown in Fig.1, and there is no Fig.1a labeled. Is this figure missing?

M1 and M2 are included in new Figure 1a. New Figure 1b is domain of the model; which is included in the revised version.

l.219: Cumulative net heat loss should have units of J m^-2 instead of W m^-2.

Thank you. We have changed the units.

Fig.2: The wind direction (panel b) often has steep downward slopes indicating counterclockwise rotation in approximate resonance with the inertial period. Is there much evidence of inertial oscillations in the simulations following these events?

The study is located in a region where the local inertial period (22.6 Hours) is close to 24 hours and thus a resonance condition between the local sea breeze system and inertial currents occur particularly during the summer months when strong prominent sea breeze cycles occur [19]. In winter this influence is reduced. We did not observe strong inertial currents in the modelling – perhaps because of the shallower depths (< 50m).

 Fig.5: Caption and figure labels are missing a few items, including whether panels c,d,g, and h are ADCP or model, U or V, and averaged over event A or B. I'm guessing panel c is ADCP U for event A, d is ADCP U for event B, and g and h are ADCP V for the two events.

Velocity vector profiles from model are included in new Figure 5. Although there are some variations in few locations between ADCP and model velocity, but mostly they are in good agreement.

l.383-384: Given the spatial inhomogeneity and large gaps between plumes shown in Figs.8 and 9, it seems odd to suggest that a single transect would be sufficient to identify DSWC.

This is not the suggestion rather the aims of the paper to examine the wider extent. The field data (ocean glider) were confined to a single transect – and the model was used to verify whether the occurrence of DSWC occurred only along a single transect or widespread over the study region. The model confirmed the latter.

Reviewer 2 Report

Review jmse-422121

The paper describes a topic which is typical of a certain geographical area, but it seems to me of  general interest for the way it is handled, i.e. the 3D modelling and the linkage of the horizontal and vertical density gradients to meteorological forcing.

I suggest a minor revision, especially to: 1) better mark all the addressed transects in the provided map or domain, 2) merge the discussion and conclusion sections, observing that the conclusion is often a repetition of what already written in the discussion.

Please check also punctuation, some typos and references in the text. 

Minor:

Lines 103-104: check the sentence

Line 157: you could add information also about the used turbulence closure model, to complete

Line 217: you refer to Figure 1a, but you have only Figure 1 in the paper. Please also show on this Figure where M1 and M2 are

Line 279: in the caption of Figure 5 I would write something like ‘modelled velocities and velocity measured from ADCP…’. Referring to the same Figure, the average cross-shore and along-shore profiles are presented only for the ADCP measurements. You could compute them also for the modelled velocities, which could be of help for a rapid qualitative comparison.

Line 301: In the caption of Figure 6 you refer to the predicted Two Rocks temperature transects. Is the transect only one, but with time-varying temperature? not clear 

Line 307: check ‘overlain’

Line 365: When you refer to the along-shelf transect extracted along the 40m depth, whose information are later shown in figure 9 and figure 13, you should mark this transect on the map (i.e. Figure 1). It would be clearer to understand where we are.

Line 375: in the caption check ‘an shore’

Line 454-455: check ‘Two Rocks transects’ and ‘obtained’

Line 485: check 'both' (maybe you refer also to [3])

Line 486: you can delete the derivative (not necessary, because you do not refer to a similar expression anywhere else)

Line 511: check dot

Lines 516 and 518: Please check through the whole manuscript ‘DSWC’. I guess it is plural, but sometimes it is addressed as singular noun. For example, see here: ‘DSWC’s were’

Conclusion: many conclusions are already written in the discussion. Please remove them and merge discussion and conclusion in a single section (if not differently required from the journal format)

Author Response

Response to reviewers: JMSE 422121 (Submitted to Journal Marine Science and Engineering)

Spatial and Temporal variability of Dense Shelf Water Cascades along the Rottnest continental shelf in south-west Australia

by Tanziha Mahjabin, Charitha Pattiaratchi, Yasha Hetzel and Ivica Janekovic.

We would like to thank and acknowledge the three reviewers and the editor for their careful reading and constructive comments on the manuscript. We believe that we have addressed the issues raised by reviewers and the proposed changes to the manuscript are detailed in this document. We trust that the reviewers and the editor will find that the suggested changes will make the manuscript to be suitable for publication.

In the following, black indicates the comments by the reviewer; blue is our response to the reviewers.

Response to Reviewer#2

Following are the revisions and response to the comments of the reviewer:

Lines 103-104: check the sentence

The sentence is edited as suggested

Line 157: you could add information also about the used turbulence closure model, to complete

Generic Length Scale model [43] was used to parameterise turbulence.

Line 217: you refer to Figure 1a, but you have only Figure 1 in the paper. Please also show on this Figure where M1 and M2 are

M1 and M2 are included in new Figure 1a. New Figure 1b is domain of the model; which is included in the revised version.

Line 279: In the caption of Figure 5 I would write something like ‘modelled velocities and velocity measured from ADCP…’. Referring to the same Figure, the average cross-shore and along-shore profiles are presented only for the ADCP measurements. You could compute them also for the modelled velocities, which could be of help for a rapid qualitative comparison.

The caption of Figure 5 was revised according to suggestion.

Velocity vector profiles from model are now included in new Figure 5. Although there are some variations in few locations between ADCP and model velocity, but mostly they are in good agreement.

Line 301: In the caption of Figure 6 you refer to the predicted Two Rocks temperature transects. Is the transect only one, but with time-varying temperature? not clear

It is same transect with varying time period in Figure 6. We noted that it is the ‘temperature transect’.

Line 307: check ‘overlain’

We have replaced overlain in the revised version.

Line 365: When you refer to the along-shelf transect extracted along the 40m depth, whose information are later shown in figure 9 and figure 13, you should mark this transect on the map (i.e. Figure 1). It would be clearer to understand where we are.

We included a red dashed line in Figure 1a show the location of the transect (it is close to the 40 m isobath).

Line 375: in the caption check ‘an shore’

It is replaced with ‘a shore’.

Line 454-455: check ‘Two Rocks transects’ and ‘obtained’

Thank you for noticing the sentence. It needed to be fixed structurally. It is changed in the revised version.

Line 485: check 'both' (maybe you refer also to [3])

We have referred another case study with reference 5; this will clarify ‘both’.

Line 486: you can delete the derivative (not necessary, because you do not refer to a similar expression anywhere else)

The derivative term is deleted according to suggestion.

Line 511: check dot

Corrected according to suggestion.

Lines 516 and 518: Please check through the whole manuscript ‘DSWC’. I guess it is plural, but sometimes it is addressed as singular noun. For example, see here: ‘DSWC’s were’

Yes, we agree with you that DSWC is plural in the most places of the manuscript. It is now consistent in the revised manuscript.

Conclusion: many conclusions are already written in the discussion. Please remove them and merge discussion and conclusion in a single section (if not differently required from the journal format)

In the Journal format, there is Conclusion as heading and this section has been retained.

Reviewer 3 Report

Review of:

Journal: JMSE

Manuscript ID: jmse-422121

Title: Spatial and Temporal variability of Dense Shelf Water Cascades along the Rottnest continental shelf in south-west Australia

Authors: Tanziha Mahjabin, Charitha Pattiaratchi, Yasha Hetzel, Ivica Janekovic

recommendation: moderate revision

General comments:

The MS deals with the observation and modeling of dense shelf water cascade at the western Australian shelf. The observations have high quality (also glider data is present), and the modeling presented is sound. The MS is, however, a little bit lengthy and therefore a bit tiring with all the descriptions of the various pathways, wind conditions and downwelling events. I wonder if this could be a bit more streamlined and shortened.

I include here below also some other points the authors should respond in the MS. Especially one point is annoying, which is how they present the vertical cross-shore sections. They cannot decide about the verse of the section, so sometimes they present them west to east, and sometimes east to west.

One more point: when the authors show the comparison between glider and modeling data, the bathymetries are completely different. Maybe it is an error, but maybe this is the results of strong smoothing in the modeling domain. In the latter case, they will have to think about something different, because the bathymetries do not seem compatible between each other.

Specific comments:

31-32: what is the role that the rivers play in this picture?

97: can you give numbers? What is the contribution of the river with respect to rain and evaporation?

99: in line 31 it was 2.5 m

103: shouldn’t it be more in autumn that this effect are more pronounced? I autumn you would have saltier and colder water along the coast…

Fig 1: in the inset it is not clear where the study area is located

136-138: please explain why you think that a hydrostatic model is adequate for describing gravity currents and deep water formation. Wouldn’t a non-hydrostatic model be more adequate?

139: I cannot find figure 1b

145: so you use sigma layers. Is the bathymetry gradient not to strong for using sigma layers? What do you do to avoid hydrostatic consistency instability

156-157: can you give a reference for the upwind 3rd order scheme?

161: no figure 1b

191: you never said what is your simulation period

195-202: I do not understand this. Why do you not use density as a proxy? Especially as you mentioned before that water temperature is higher along the coast. Therefore, you are following a warmer vein? This is counter intuitive.

228-9: confusing. Why don’t you just show UTC or local time?

Fig 4: very strange. The bathymetry looks completely different. Are you sure the transect in the model and for the glider are the same? Did you have to do a heavy smoothing in the model? The two domain look not comparable to me.

259-260: I would be not so sure looking at figure 5

282-284: not clear if what you show in panels c,d and g,h is from model or ADCP. To me the comparison between a,b and e,f look quite different.

Fig 6: with respect to figure 4 you have changed the direction of the section. Not a good idea. This will only confuse the reader. I propose to invert figure 4, because when looking at the horizontal setting figure 6 is closer to what one would expect. I would also recommend to indicate these days during which there is strong wind forcing.

363: figure 8c or 8d?

375: from landward doesn’t make sense. You probably want to say from land.

Fig 10: the transects should be shown in an overview (horizontal) map. It is difficult to orient only with lat/lon values.

405: if the wind is blowing to the west, then there should be no onshore component… sorry, I have understood now. It seems that you say that the wind blows from north to the west direction. Maybe you should reformulate: northwesterly winds, or winds from north and west…

Fig 15: now the verse of the section is again reversed. This is not acceptable.

Author Response

Response to reviewers: JMSE 422121 (Submitted to Journal Marine Science and Engineering)

Spatial and Temporal variability of Dense Shelf Water Cascades along the Rottnest continental shelf in south-west Australia

by Tanziha Mahjabin, Charitha Pattiaratchi, Yasha Hetzel and Ivica Janekovic.

We would like to thank and acknowledge the three reviewers and the editor for their careful reading and constructive comments on the manuscript. We believe that we have addressed the issues raised by reviewers and the proposed changes to the manuscript are detailed in this document. We trust that the reviewers and the editor will find that the suggested changes will make the manuscript to be suitable for publication.

In the following, black indicates the comments by the reviewer; blue is our response to the reviewers.

Response to Reviewer #3

Following are the revisions and response to the comments that the reviewer pointed out:

31-32: what is the role that the rivers play in this picture?

The study region is a low rainfall region (mean annual rainfall ~0.60m) The main river system: Swan River discharge is very small and does not have a major influence on the dynamics of the region. Its influence is <5 km from the river mouth at Fremantle.

97: can you give numbers? What is the contribution of the river with respect to rain and evaporation?

The mean annual discharge of the river is < 8 m3s-1 with the main flow occurring in June/July/August. The annual rainfall is 0.6m whilst evaporation is 2.5m. Thus the annual evaporation is a factor 4 larger than the annual rainfall.

99: in line 31 it was 2.5 m

We made it consistent as 2.5 m

103: shouldn’t it be more in autumn that this effect are more pronounced? I autumn you would have saltier and colder water along the coast…

Yes, we agree but believe the reviewer has not considered that the study is located in southern hemisphere. Here, May and June are represents late autumn and early winter period (mentioned in line 79). That’s why the density gradient is at its maximum and DSWC are observed frequently during this period.

Fig 1: in the inset it is not clear where the study area is located

The dashed line in Figure 1a shows the model domain which is Perth region. A box is included to show the region in the inset as well. Also we included Figure 1b where the child domain is showing the exact location of the study area.

136-138: please explain why you think that a hydrostatic model is adequate for describing gravity currents and deep water formation. Wouldn’t a non-hydrostatic model be more adequate?

The ROMS model using the hydrostatic assumption has been shown in numerous studies to be appropriate for the coastal ocean at this scale and is more computationally efficient. In particular, similar model setups have been used successfully to investigate dense water flows [44, 45, 46, 47]. It has also been shown that the hydrostatic assumption only causes issues near the plume head [48]. Since we are not investigating dynamics at the plume head, our model is appropriate for this paper. For more detailed studies at finer spatial scales, a non-hydrostatic model may be required but at this time the non-hydrostatic ROMS model is still in development.

We have edited the methods section as follows to more clearly justify this assumption (line 166-169):

As shown in other similar studies [44, 45, 46, 47]  this configuration includes non-linear dynamics whilst the hydrostatic approximation used is not a major limitation, because the horizontal scales of motion are larger than vertical, and effects have been shown to be limited to plume head dynamics for extreme cases [48] which is not the focus here.”

Roveri et al. 2014 [45], used the ROMS to model DSWC and justified the approach as follows:

“The Regional Ocean Modeling System model uses a stretched terrain following coordinates system in the vertical, to better adapt the grid to variable steep topography. The 30 vertical sigma levels were stretched to have the highest resolution near the sea floor. In this configuration, all non-linear dynamics have been resolved and the hydrostatic approximation used is not a major limitation, because the horizontal scales of motion are in any case much larger than the vertical ones. Moreover in our case we could have an underestimation of the generated current field; this means that our speculation on their effects is “/a fortiori/” good. Ulses and others (2008) already applied a hydrostatic model to investigate the dynamics of cascading in the Gulf of Lion basing their methodology on the work of Heggelund and others (2004) that showed how the approximation has a significant effect only on the plume head. Generally, in the Regional Ocean Modeling System, the solutions of the Navier-Stokes equations are computed over a relative wide grid not suitable for considering the effects of small-scale turbulence that for this reason has to be parametrized via a closure term; in our case we used the Generic Length Scale (Haidvogel and others, 2008 with a K-omega parametrization and different types of stability function and horizontal smoothing of buoyancy/shear.”

139: I cannot find figure 1b

Figure 1b is domain of the model; which is included in the revised version.

145: so you use sigma layers. Is the bathymetry gradient not too strong for using sigma layers? What do you do to avoid hydrostatic consistency instability?

See previous comment for line 136-138. The bathymetry gradients are not as strong as the studies cited above that have successfully modelled DSWC, and some smoothing has been done in order to minimise errors.

The following has been added to the text at Line 151 for better explanation:

Final bathymetry was minimally smoothed using a linear programming approach [37] to suppress horizontal pressure gradient errors.”

156-157: can you give a reference for the upwind 3rd order scheme?

See line 163: “…while for the parent system the upwind third-order scheme was used [42]”

161: no figure 1b

Figure 1b is domain of the model; which is included in the revised version.

191: you never said what is your simulation period

Simulation period for the model is 2015-present (Line 137). And analysis period for this study is May-June 2016 (Line 139).

195-202: I do not understand this. Why do you not use density as a proxy? Especially as you mentioned before that water temperature is higher along the coast. Therefore, you are following a warmer vein? This is counter intuitive.

No we are following colder water. In southern hemisphere, May and June represent late autumn and early winter period (mentioned in line 79); which results in colder water in winter in the nearshore compared to summer. That’s why the density gradient is at its maximum and DSWC are observed frequently. So we can use temperature as proxy, as density is defined by the changes in temperature.

228-9: confusing. Why don’t you just show UTC or local time?

All time are in UTC. But mentioning the local time was crucial here as it is representing 12 pm midday; when the heat flux was at peak.

Fig 4: very strange. The bathymetry looks completely different. Are you sure the transect in the model and for the glider are the same? Did you have to do a heavy smoothing in the model? The two domain look not comparable to me.

Yes it is partially correct that there was smoothing in the model (as well as availability of high resolution bathymetry data). Also model does not resolve all local features of bathymetry. But as we can see the pattern of DSWC formation is similar in both cases; which is our main goal here.

We have reversed the transects to match with model generated transects according to suggestion.

259-260: I would be not so sure looking at figure 5

Figure 5 has been updated

282-284: not clear if what you show in panels c,d and g,h is from model or ADCP. To me the comparison between a,b and e,f look quite different.

Velocity vector profiles from model are included in Figure 5. Although there are some variations in few locations between ADCP and model velocity, but mostly they are in good agreement.

Fig 6: with respect to figure 4 you have changed the direction of the section. Not a good idea. This will only confuse the reader. I propose to invert figure 4, because when looking at the horizontal setting figure 6 is closer to what one would expect. I would also recommend to indicate these days during which there is strong wind forcing.

We have reversed Figure 4 and 15 to match with model generated transects. We can mention where the strong wind is, but the direction also main factor here. Because sometimes wind could be strong with offshore direction wind, that will not enhance DSWC. On the other hand, DSWC would be enhanced with onshore strong wind. So it is preferable to look at Figure 3; that will give information on both with speed and direction.

363: figure 8c or 8d?

Yes it is supposed to be Figure 8d. Thank you for the comment.

375: from landward doesn’t make sense. You probably want to say from land.

It is replaced with ‘from land’.

Fig 10: the transects should be shown in an overview (horizontal) map. It is difficult to orient only with lat/lon values.

The transects are shown according to P1-P5 flow pathways which are shown on the map in Figure 8. I have edited the Figure 10 caption according to that. Also same way I edited Figure 14 caption mentioning about Figure 12.

405: if the wind is blowing to the west, then there should be no onshore component… sorry, I have understood now. It seems that you say that the wind blows from north to the west direction. Maybe you should reformulate: northwesterly winds, or winds from north and west…

We have edited it as ‘winds from north and west’ in the revised manuscript.

Fig 15: now the verse of the section is again reversed. This is not acceptable.

We have reversed the transects to match with model generated transects according to suggestion.
